# Two-dimensional high-throughput on-cell screening of immunoglobulins against broad antigen repertoires
Yakov A. Lomakin [1] ✉, Leyla A. Ovchinnikova[1], Stanislav S. Terekhov[1], Samir S. Dzhelad[1], Igor Yaroshevich[1,2], Ilgar Mamedov[1], Anastasia Smirnova[1], Tatiana Grigoreva[1], Igor E. Eliseev[1], Ioanna N. Filimonova[1], Yuliana A. Mokrushina[1], Victoria Abrikosova[1], Maria P. Rubtsova[1,3], Nikita N. Kostin [1], Maria A. Simonova[1], Tatiana V. Bobik[1], Natalia L. Aleshenko[4], Alexander I. Alekhin[4], Vitali M. Boitsov[5], Hongkai Zhang[6], Ivan V. Smirnov[1,7], Yuri P. Rubtsov [1,8] & Alexander G. Gabibov [1,9,10] ✉

Identifying high-affinity antibodies in human serum is challenging due to extremely low number of circulating B cells specific to the desired antigens. Delays caused by a lack of information on the immunogenic proteins of viral origin hamper the development of therapeutic antibodies. We propose an efficient approach allowing for enrichment of high-affinity antibodies against pathogen proteins with simultaneous epitope mapping, even in the absence of structural information about the pathogenic immunogens. To screen therapeutic antibodies from blood of recovered donors, only pathogen transcriptome is required to design an antigen polypeptide library, representing pathogen proteins, exposed on the bacteriophage surface. We developed a two-dimensional screening approach enriching lentiviral immunoglobulin libraries from the convalescent or vaccinated donors against bacteriophage library expressing the overlapping set of polypeptides covering the spike protein of SARS-CoV-2. This platform is suitable for pathogen-specific immunoglobulin enrichment and allows high-throughput selection of therapeutic human antibodies.

Recently, the development of therapeutic antibodies has largely depended on the isolation of B-cell clones with the required specificity. Serum from convalescent patients serves as the main source of highly effective neutralizing antibodies. However, an extremely low fraction of antigen-specific B cells in circulating blood complicates isolating and characterizing these potentially therapeutic antibodies. Developing therapeutic drugs and vaccines against coronavirus disease 2019 (COVID-19) caused by severe acute respiratory syndrome coronavirus 2 (SARS-CoV-2) required extensive studies of SARS-CoV-2-specific humoral immunity driven by the secretion of potent anti-viral antibodies. Several therapeutic monoclonal antibodies (mAbs) have received approval from regulators and have shown to be promising in treating immunocompromised and elder individuals[1].

Purified recombinant biotin-labeled full-length SARS-CoV-2 spike protein (S protein), its truncated forms, or even single domains, were used to isolate and characterize specific immunoglobulins and the corresponding B cells from convalescent blood[2,3]. This conservative approach relies on biotinylated antigens (Ags) and fluorescence-activated cell sorting (FACS) of antigen-specific (Ag-specific) B cells. Accessing the functionality of identified immunoglobulins requires the production of hundreds of full-length recombinant mAbs with subsequent routine confirmation of affinity and neutralizing properties[4–6]. This is further complicated by the need of determining the exact immunogen. In the case of COVID-19, the immunogenic linear epitopes of the S protein were revealed by the peptide microarray covering full-length S protein[7]. Unfortunately, peptides shorter than 12 a.a. were shown to be ineffective in binding with the neutralizing

[1]Shemyakin-Ovchinnikov Institute of Bioorganic Chemistry RAS, 117997 Moscow, Russia. [2]Department of Biophysics, Faculty of Biology, Lomonosov Moscow State University, 119991 Moscow, Russia. [3]Chemistry Department, Lomonosov Moscow State University, 119991 Moscow, Russia. [4]Federal State Budgetary Scientific Institution «Petrovsky National Research Centre of Surgery» (FSBSI «Petrovsky NRCS»), Moscow, Russia. [5]Saint Petersburg National Research Academic University of the Russian Academy of Sciences, 194021 Saint Petersburg, Russia. [6]College of Life Science, Nankai University, Tianjin, People's Republic of China. [7]Endocrinology Research Centre, Ministry of Health of Russia, 117036 Moscow, Russia. [8]Blokhin National Medical Research Center of Oncology, Ministry of Health, Moscow, Russia. [9]Faculty of Biology and Biotechnology, HSE University, 101000 Moscow, Russia. [10]Faculty of Medicine, Lomonosov Moscow State University, 119192 Moscow, Russia. ✉e-mail: yasha.l@bk.ru; gabibov@gmail.com

antibodies. The majority of SARS-CoV-2-neutralizing antibodies recognize conformational epitopes within the receptor-binding domain (RBD), which are likely attributed to its unique conformational flexibility[8].

After a population of Ag-specific B cells is obtained, single-cell RNA sequencing (scRNA-seq) can be effectively applied to study these cells[9,10]. The main advantage of scRNA-seq is the ability to identify the sequences of individual B-cell Receptors (BCRs) and characterize phenotypic markers of each analyzed B cell within a sample. Moreover, pre-exposure of analyzed cells with several biotinylated and barcoded Ags appended to classic scRNA-seq allow to study single Ag-specific B cells with various specificity in a high-throughput manner[11]. Despite the promise of using scRNA-seq to identify Ag-specific Abs, the limitation in the number of analyzed cells (up to 20000 per assay), high cost per sample, and the necessity to obtain recombinant antigens in the first instance, complicates the identification of rare therapeutic Ag-specific Abs with an a priori unknown specificity. Alternatively, therapeutic antibodies against most new disease-associated biomolecules can be obtained via phenotypic drug discovery approach[12]. This cell-based phenotypic screening theoretically enables identifying novel drugs without understanding a complex mechanism of pathogenesis. However, hit validation and target identification are not trivial for immunoglobulins[13,14]. Alternatively, phage immunoprecipitation sequencing (PhIP-seq) is applied to systematically screen for antibodies against hundreds of thousands of different antigens simultaneously[15,16]. Garrett et al. constructed a phage-based deep mutational scanning library comprising all possible identified S-protein single a.a. mutations in the format of phage-anchored 31 a.a. long peptide fragments[17]. Profiling the pattern of antibody binding with 24820 unique peptides across the S protein in the plasma of COVID-19 patients uncovered a spectrum of mutant proteins demonstrating a reduced binding with antibodies. Person-to-person variability in the effect of mutations within immunodominant epitopes was also documented. Unfortunately, the majority of antibodies recognizing RBD that are capable of neutralizing the virus bind to conformational epitopes not present in the 31-mer SARS-CoV-2 peptide library used in the study.

To overcome the lack of conformational epitopes in the peptide library Ravichandran et al. constructed SARS-CoV-2 fragment phage display library encoding polyproteins ranging from 18 to 500 a.a. long[18]. This library adsorbed >92% of SARS-CoV-2 virion-specific serum antibodies in patients with COVID-19 providing proof of concept for using the SARS-CoV-2 phage library for antibody epitope repertoire analyses. Nevertheless, the proposed methodology reveals only the epitope profile and antibody response to the analyzed epitopes, while the sequences of epitope-specific mAbs could not be identified.

Another challenge in studying antigen-specific antibodies is proper VH-VL pairing. Despite the great number of antigen-specific and even therapeutic mAbs obtained after enriching combinatorial paired VH-VL libraries[19–24], it has been demonstrated that the native chain pairing of antibodies from immunized sources can facilitate the identification of a higher number of functional clones[25]. For COVID-19, heavy-light swaps between four analyzed SARS-CoV-2-neutralizing antibodies were shown to substantially reduce the neutralization for the non-native heavy-light combinations[26]. Meanwhile, a small sample size still did not resolve the question whether there were antibodies containing heavy chains paired with different light chains with similar antigen-binding properties. On the other hand, a promising approach involves screening public light chains that have been paired with antigen-specific VH, or employing yeast Fab display to randomly combine the aforementioned VH with donor VL[27]. Here, along with the high-throughput platform for antigen-specific B-cell screening, we aimed to confirm that functional neutralizing antibodies with the same heavy but different light chains can be obtained from hundreds of different immunoglobulin clones from human serum.

Mutations in the SARS-CoV-2 genome, and in the RBD domain in particular, accumulate rapidly to protect the virus from neutralizing antibodies. This tendency requires rapid protocols for developing new antibodies against SARS-CoV-2 antigens. In this paper, we present a comprehensive platform for producing high-performance therapeutic antibodies from an antibody library, that potentially will not require pre-determined information on the target antigen structure. It combines selection with a library of antigens presented on a bacteriophage, lentiviral antibody display, next-generation sequencing, and bioinformatics analysis. This platform enables the screening of therapeutic antibodies with desired but a priori unknown specificity.

## Results
### Study design
This study proves the concept that virus-neutralizing and pathogen-specific antibodies can be discovered in the human sera even if the exact structures of the immunogens are unknown. If the genome of the pathogen was sequenced, one could split its predicted proteome to 50–300 a.a. over-lapping fragments and express as a phage display library. The obtained phage repertoire will carry potential immunogenic epitopes of the pathogen (Fig. 1). This antigen phage library can be screened against the reporter cell line expressing (in scFv-Fc or Fab format) the lentiviral human antibody library of recombinant analogs of BCRs derived from circulating B cells. Immunoglobulin-expressing cells that bind antigen-decorated phages can be FACS sorted to obtain sequences of pathogen-specific Ig. The advantage of the proposed strategy is that there is no need in individual expression, purification, and obtaining the panel of pathogen-derived proteins in the soluble form. This strategy significantly reduces the time required to identify effective Ag-specific Ig. It takes less than 2 weeks to construct an antigen library (see protocol pipeline in Fig. 1). Moreover, the proposed platform allows Ag-specific antibody enrichment to be efficiently performed prior to the immunogen identification.

### SARS-CoV-2 spike protein phage library
In this work, we used SARS-CoV-2 Spike protein as a model viral protein for screening of the high-affinity virus-specific and neutralizing antibodies. Normally, Spike protein is required for attaching to the host cell and penetrating into it. The spike consists of three domains, where RBD is the main target for neutralizing antibodies preventing the binding of RBD to ACE2 (Angiotensin-converting enzyme 2). Most of the identified monoclonal SARS-CoV-2-neutralizing antibodies target conformational epitopes located within this receptor-binding domain (N331-V524). We designed a small Phage Antigen Library (PhAgL) containing full-size RBD (Spike$_{331-530}$) and its immunodominant fragments: Spike$_{354-519}$, Spike$_{358-410}$, Spike$_{539-583}$, Spike$_{795-839}$ (Fig. 2). Spike$_{354-519}$ was chosen due to the existence of mono-clonal SARS-CoV-2/SARS-CoV neutralizing antibodies recognizing conserved epitopes in the range of residues Y369-H519[28]. Spike$_{539-583}$ and Spike$_{795-839}$ were selected as it was previously shown that antibodies targeting immunodominant regions T553-A570 and K809-K825 located outside the receptor-binding domain significantly alter virus neutralization capacities[29]. Spike$_{358-410}$ was included as a linear antigen potentially inducing non-neutralizing antibodies[30]. Oligonucleotide sequences encoding the corresponding polypeptides were cloned into fd bacteriophage in frame with p3 coat protein and 3xFLAG epitope[31].

### Lentiviral antibody library construction
Pseudovirus neutralization assays confirmed that convalescent/immunized sera potently neutralized SARS-CoV-2 due to the presence of immunoglobulins recognizing the spike protein (Supplementary Table 1). B cells were isolated from the blood of individual donors using B-cell enrichment kit. Biotinylated recombinant Spike protein was used to capture Ag-specific B cells for preliminary enrichment using FACS. The resulting pools of B cells were used to construct the randomly paired VH-VL libraries. This resulted in high average number of individual clones in all five constructed antibody libraries (Supplementary Table 2). We identified $93 \pm 63$ variants of Ig heavy variable ($V_H$) and $174 \pm 111$ variants of Ig light variable ($V_L$) fragments (mean ± SD) per each library (out of 5, Supplementary Table 2). Therefore, provided that any heavy chain can be paired with any light chain, the theoretically possible diversity of the obtained libraries ranged from $3.5 \times 10^3$ to $5.1 \times 10^4$ individual VH-VL clones. All obtained VH-VL

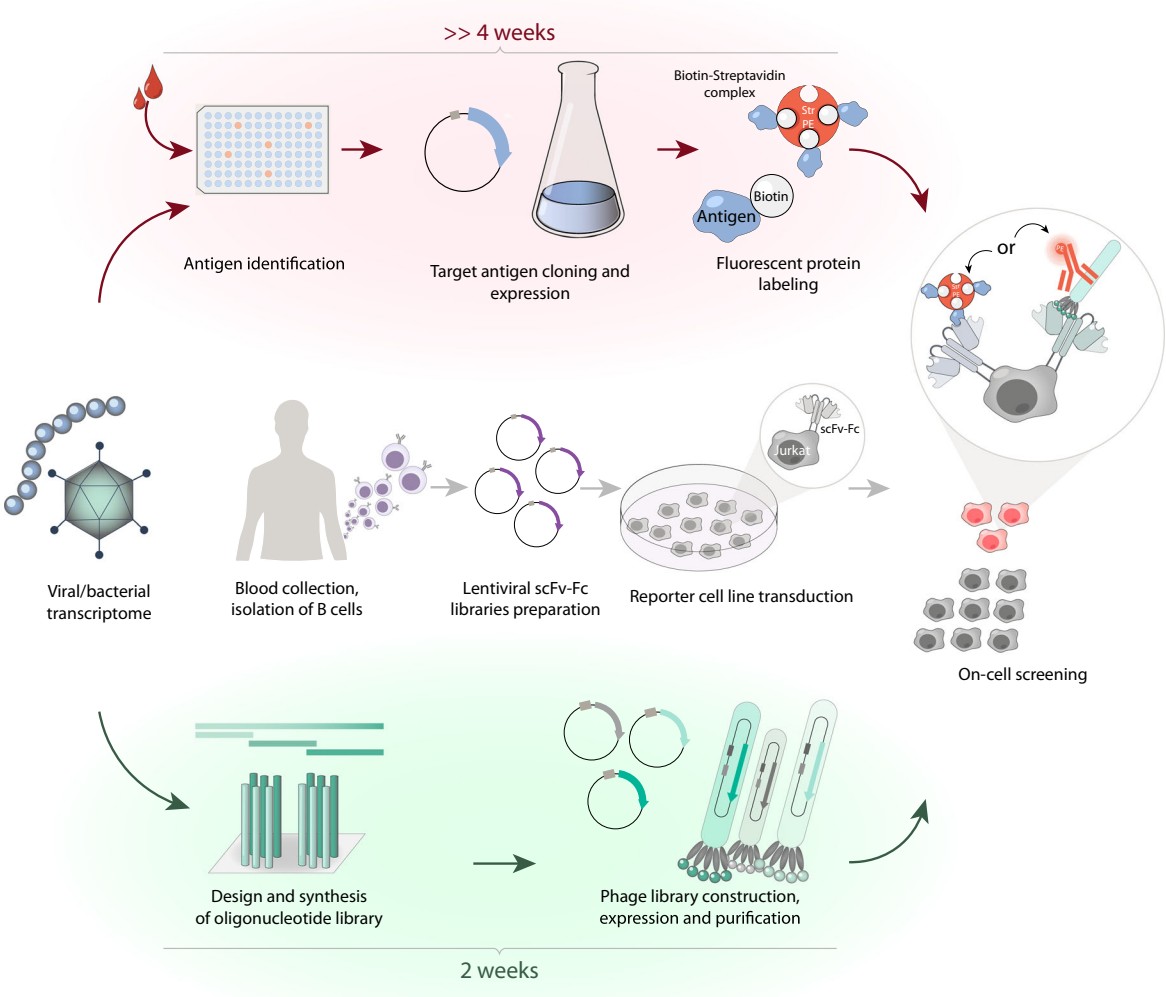

**Fig. 1 | Scheme outlining the antigen-specific B cell isolation strategy using the phage antigen library (PhAgL) and flow cytometry sorting.** Antibody library, based on Ig repertoire from recovered donors, represents transgenic eukaryotic cells, i.e., Jurkat cells, expressing membrane-tethered BCRs in a scFv-Fc format. Classic isolation of Ag-specific mAbs implies preliminary targeted Ag identification and its production in soluble form (shaded red), while PhAgL screening requires only the pathogen transcriptome for designing the Ag library (shaded green).

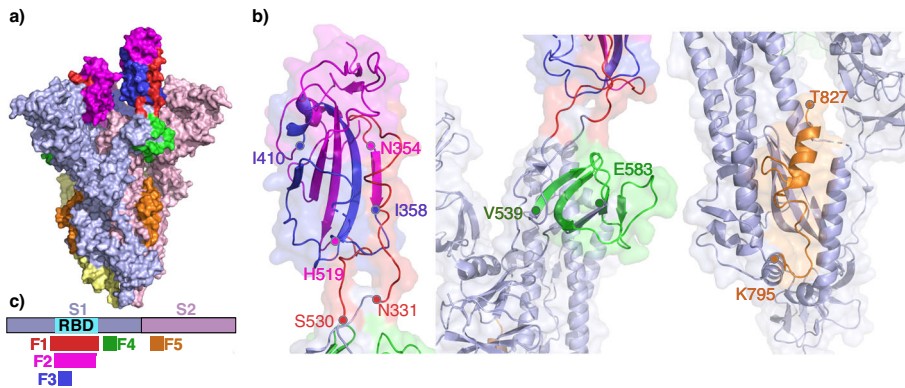

**Fig. 2 | Design of Spike phage antigen library (PhAgL).** Structure of the trimer of S-protein: **a** 7fcd structure is used, subunit colors are: pale yellow, pale cyan, and pale pink, and **b** corresponding antigen fragments: F1 (red-magenta-blue, N331-S530), F2 (magenta-blue, N354-H519), F3 (blue, I358-I410), F4 (green, V539-E583), F5 (orange, K795-D839, T827 marked as a last resolved residue in the motif). **c** S-protein primary structure with the corresponding fragments.

libraries were cloned into lentiviral pLV2 vector in a single-chain variable fragment (scFv) format[32]. Thus, the scFv fused with the constant domain of antibody (Fc) was linked to a membrane-spanning domain of the PDGFR (platelet-derived growth factor receptor) by a flexible linker. In this case, the antibodies were expressed as dimers on the plasma membrane. Importantly, the antigen-binding sites were exposed on the cell surface, facing the solvent.

Since several viral particles can simultaneously infect single eukaryotic cell[33], several different scFvs can be expressed on the surface of a single cell. Therefore, irrelevant accompanying VH-VL combinations should be reduced during biopanning. For a library expressed in the cells as monoclonal scFv, it is necessary to optimize the ratio of viral particles to cells. As the ratio of cells transduced with the VSV-G-pseudotyped retroviral vectors correlates with the result predicted by Poisson distribution[34], we calculated the probability of the cell infected by two and more lentiviral particles (Supplementary Fig. 1). To obtain lentiviral libraries, containing more than 80% of cells infected with single viral particle, a fraction of infected cells should be less than 30%. Notably, the chance of multiple viral integration into the genome of a single cell is much smaller than the theoretical estimates[33,35]. This means that less than 30% transduction ratio ensures the generation of cells, expressing individual VH-VL combinations. Therefore, we performed the lentiviral titration before each round of biopanning for each library and obtained libraries with ~10–30% of Fc-positive cells.

### Ig library biopanning against recombinant RBD or PhAgL

Lentiviral antibody display libraries were screened independently for binding to SARS-CoV-2 fragments using FACS. For biopanning with recombinant RBD (rRBD) molecule, we constructed RBD-streptavidin tetramers conjugated with two different fluorochromes and enriched the population of cells stained with both RBD tetramers (Fig. 3a). For biopanning with PhAgL, we used fd bacteriophages exposing fragments of Spike fused with 3xFLAG epitope. In this case, we defined the population of cells staining positive for FLAG epitope as spike-specific.

Regardless of the enrichment protocol, each IgG library was subjected to the same number of biopanning rounds with recombinant RBD or PhAgL to obtain up to 95–99% of positive Ag-specific clones (Fig. 3a). We performed 2 rounds of biopanning for the libraries from the vaccinated donors and 3 rounds of biopanning for those from the recovered donors. Then the binding of the cell pools enriched by both protocols with the recombinant RBD molecule or PhAgL was compared under identical conditions. Both protocols resulted in similar enrichment efficiency. Additionally, three rounds of biopanning with scFv library from a healthy donor were done to demonstrate a lack of significant enrichment of RBD-positive clones.

Next, the ability of RBD-enriched libraries was assessed in competition tests with human ACE2. In experiments when the pre-mixed RBD and ACE2 interactions were disrupted by adding enriched RBD-binding cells, only 5-fold molar excess of ACE2 allowed distinguishing between ACE2 competing and ACE2 non-competing RBD-binding clones (Fig. 3b, Supplementary Fig. 2). Therefore, we sorted four fractions of RBD-positive clones from each library depending on their competition with recombinant ACE2: ACE−−, ACE−, ACE+, ACE++ fractions.

As the enrichment efficiency with PhAgL was compared with that on recombinant RBD molecule, PhAgL was screened for the most immunodominant spike fragment. To this end, PhAgL-enriched and RBD-enriched scFv libraries were stained with individual phages exposing only one variant of spike fragment. Binding was attributed exclusively to binding with the longest spike fragment - Spike$_{331-530}$ (Fig. 3c, Supplementary Fig. 3).

### NGS data analysis, candidate IgG selection, and validation

VH-VL sequences from the enriched and initial libraries were analyzed using NGS. More than 10,000 paired reads per each enriched and >200,000 paired reads per each initial library of extracted VH and VL sequences were obtained. Precise analysis of accurate VH-VL pairing was performed using our internal analysis pipeline. Only the sequences covered by at least two reads were included into the downstream analysis. The reliability of the immunoglobulin chains combination was evaluated based on the harmonized mean of the specificity metric of the VH amplicons for the paired VL and vice versa.

As we expected, heavy and light chains were randomly paired in all initial scFv libraries prior to biopanning (Fig. 4a). Each heavy chain was paired with at least 5–10 interchangeable light chains. All variants of biopanning led to identifying 1–3 dominant VH-VL combinations with each heavy chain from the enriched pools. The enrichment steps resulted in a reduction in the diversity of antigen-binding clones and the number of paired VH-VL combinations (Fig. 4b, Supplementary Fig. 4). Biopanning with either PhAgL and recombinant RBD molecule resulted in the prevalence of 12 and 12.4 individual clones respectively, while further enrichment based on competition with ACE2 decreased the number of individual clones to 7.6–10.2 per each $V_H$ (for the 90% coverage).

Finally, 23 most represented VH-VL combinations from all the enriched libraries (Supplementary Data 1) were selected for further analysis. For the majority of heavy chains association with at least several light chains was observed. In contrast, only one light chain from the pool of selected antibodies was paired with two different heavy chains (Supplementary Data 1).

Selected VH-VL combinations were then expressed as full-size human IgGs by co-transfecting HEK 293-F cells with heavy and light chain expression vectors. These vectors drive the production of a corresponding antibody with constant IgG1, kappa or lambda domains. Purified antibodies were assessed for binding to the rRBD by ELISA (Supplementary Fig. 5). The results indicated that our approach was effective in identifying the antigen-specific antibodies, preserving their functionality in full-size human IgG format. Only three out of 23 expressed clones demonstrated weak binding to EC$_{50}$ > 0.2 nM. Thirteen clones had EC$_{50}$ < 10 pM. Next, we performed phage ELISA using obtained monoclonal IgGs to validate the recognition capability of the Spike$_{331-530}$ fragment displayed on fd phage (Supplementary Fig. 6). Our aim was to determine if it could be recognized by antibodies other than those present in scFv-Fc format exposed on the Jurkat cell surface. Despite one of the three RBD-specific antibodies showing decreased binding with Spike$_{331-530}$ displayed on fd phage compared to eukaryotic rRBD, it was observed that all tested antibodies exhibited statistically significant binding to phage Spike$_{331-530}$ in contrast to irrelevant phages and IgG.

Pseudovirus neutralization assays showed three orders of magnitude differences in IC$_{50}$ values for the analyzed panel of antibodies (Fig. 5a, Table 1). Weak correlation was observed between the efficiency of RBD binding (EC$_{50}$) and pseudovirus neutralization (IC$_{50}$) by isolated mAbs (Supplementary Fig. 7). Hence, epitope specificity, rather than the affinity of the antibody's itself, has a more pronounced impact on the efficacy of neutralizing antibodies. According to the clone enrichment based on NGS analysis, potent neutralizing antibodies were found among RBD-binders, both competing and non-competing with ACE2 (Fig. 5b, c and Supplementary Fig. 8). It confirms that antibody neutralization is not simply driven by blocking the ACE2 binding site on the RBD. According to neutralizing and RBD-binding efficiency, we have identified four antibody categories: weak (IC$_{50}$ > 100 nM; EC$_{50}$ > 10000 ng/mL), mild (20 < IC$_{50}$ < 100 nM; 10000 < EC$_{50}$ < 10000 ng/mL), moderate (1 < IC$_{50}$ < 20 nM; 50 < EC$_{50}$ < 1000 ng/mL), and high (IC$_{50}$ < 1 nM; EC$_{50}$ < 50 ng/mL) (Fig. 5a). According to the surface plasmon resonance (SPR) analysis, the best identified universal neutralizing antibody Vac-3.1, demonstrates picomolar affinity towards RBD of Wuhan, Alpha, Gamma, and Omicron strains, while showing little to no interaction with the Delta RBD (Table 2). This neutralizing antibody Vac-3.1 does not compete with ACE2 or the previously published neutralizing antibody P4A1[36] for RBD-binding sites (Supplementary Fig. 9). Combining these observations with the Vac-3-1 antibody's capacity to neutralize the Wuhan and Omicron, but to a

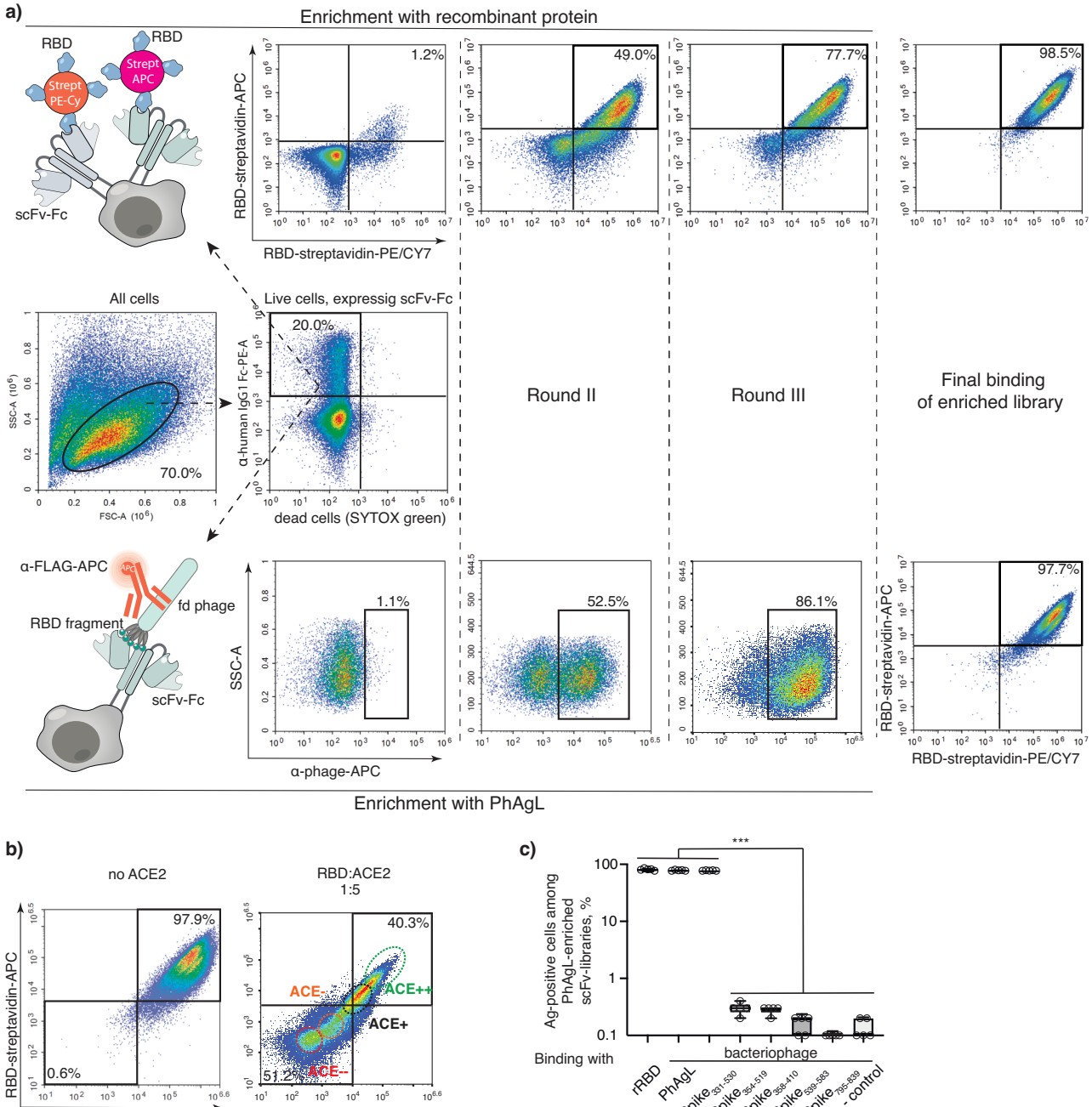

**Fig. 3 | Isolation of RBD-specific mAbs. a** Representative density FACS plots and gating strategy for cells sorted based on RBD binding of Jurkat cells transduced with scFv library. RBD-specific cells were sorted from populations enriched using bio-panning: live and dead cell separation was based on SYTOX Green staining. Additional staining with human IgG1 Fc-PE was performed to mark the infected cells producing membrane-anchored scFv-Fc. RBD-specific cells were enriched and sorted separately based on binding with recombinant RBD molecule (upper panel) and PhAgL (peptide spike library exposed on the bacteriophage surface) (lower panel). Right far panels represent the control binding with recombinant RBD in the final scFv libraries, which were enriched using two different methods.

**b** Representative FACS plots and gating strategy for sorting cells expressing RBD-specific Ig showing different competition with ACE2 in RBD-binding: ACE++ not displaced by ACE2; ACE+ partially displaced by ACE2; ACE− displaced by ACE2; ACE−− completely displaced by ACE2. **c** The percentage of Ag-specific cells among PhAgL-enriched scFv libraries towards designated antigens: recombinant RBD (rRBD), individual phages exposing spike fragments (Spike$_{331-530}$, Spike$_{354-519}$, Spike$_{358-410}$, Spike$_{539-583}$, Spike$_{795-839}$), PhAgL (equimolar ratio of these phages) and phage exposing irrelevant peptide (- control). Data are mean ± SD ($n = 5$ independent libraries). Two-tailed unpaired $t$-test was used to determine the statistical significance of values obtained for the binding of different antigens. *** $P < 0.0005$.

much lesser extent Delta strains, we can infer that Delta mutation L452R dramatically affects Vac-3.1 binding with RBD (Supplementary Fig. 10). Hence, the binding site of Vac-3.1 is presumably placed close to the L452, oppositely to the ACE2 binding site.

Based on the functional properties of the selected antibodies, it is clear that enrichment with PhAgL is not inferior to traditional enrichment with recombinant proteins. It preserves the diversity of potent binders and allows

selecting the best neutralizing antibodies with comparable efficiency (Fig. 5b, c and Supplementary Fig. 8).

## Computational analysis of immunoglobulin repertoires of SARS-CoV2-neutralizing antibodies

The necessity of highly efficient library enrichment prior to NGS analysis should be highlighted. Deep sequencing of primary enriched pools of

**Fig. 4 | Antibody enrichment data. a** Sankey diagrams depicting match scores in analyzed samples. Nodes correspond to the heavy ($V_H$) on the left and light ($V_L$) on the right IG chains. Representative set consists of initial library and enriched libraries: PhAgL, RBD, ACE+, ACE++, ACE−, ACE−−. The size of the flow connecting the nodes is proportional to the number of $V_H$-$V_L$ matches. The size of the node is proportional to the sum of the flows connected to it. The data filtration procedures are provided in the chapter Computational analysis of "Methods" section. **b** The number of dominant $V_H$-$V_L$ combinations, covering the designated portion of identified sequences in enriched libraries. Mean represented by a line, SD depicted as a shaded area.

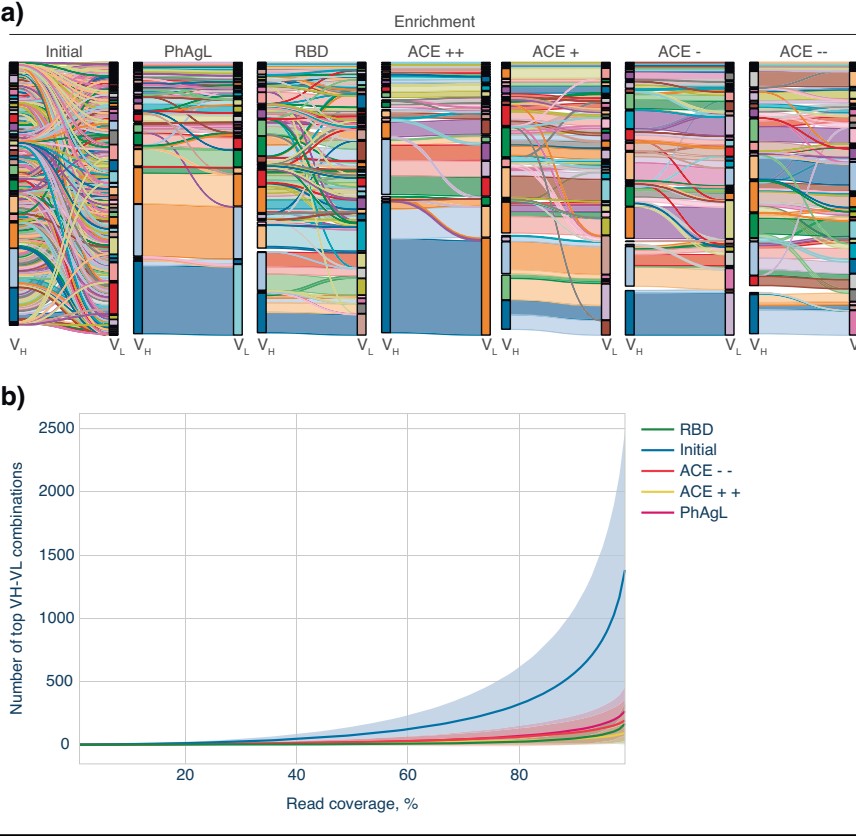

clones or even pools after initial panning rounds would likely reveal a low proportion of bona fide functional antigen-specific clones. Therefore, we analyzed $V_H$ and $V_L$ gene repertoires only following the verification of the virus-neutralizing capacity of the selected antibodies. Totally 16 functional VH-VL combinations derived from different 10 heavy chains and 16 light chains were identified (Supplementary Fig. 11, Supplementary Data 1). Discovered antibodies had similar features with the previously described SARS-CoV2-neutralizing IgGs from both vaccinated and recovered donors[37]: (i) VH-VL germline pairs (Supplementary Fig. 11), of our antibodies correspond to the pairs identified previously (i.e., none of our antibodies have a VH-VL pair that was not described before); (ii) CDR3 length (Supplementary Fig. 12a) of our antibodies matches with the IGH CDR3 length distribution of described antibodies; (iii) germline distribution (Supplementary Fig. 12c) are consistent with published data in terms of frequency of V gene usage. Virus-neutralizing sequences of $V_H$ gene segments are characterized by preferential usage of the Ig heavy chain variable region 3 (IGHV3) subfamily.

Thereafter, light chains were processed by the multiplex protocol; kappa and lambda isotypes were extracted. Both chains were present in the extracted repertoire of the light chains. The equimolar ratio of light kappa and lambda chains in the resulting VH-VL amplicons was typical of the constructed scFv libraries. The selected antibodies also have 8 variants of the kappa chain and 8 variants of the lambda chain (Supplementary Fig. 12d, Supplementary Data 1). The VH-VL combinations having the same heavy chain were functional with different light chains if only they belonged to the same isotype. Moreover, all 6 different discovered pairs of VH-VL combinations with the same heavy chain were functional only with the light chains belonging to the same Ig light family gene. Among the neutralizing antibodies, the observed CDR3 length of IGK/IGL was shorter than that reported before[37] but this could be due to the limited number of discovered light chains. 10 out of 16 functional light chains have CDR3 shorter than 11 a.a. (Supplementary Fig. 12b). $V_k$ gene segments from virus-neutralizing antibodies preferentially use the Ig kappa chain variable region 1 (IGKV1) and 3 (IGKV3) subfamily genes. $V_l$ gene segments from virus-neutralizing sequences preferentially use the Ig lambda chain variable region 3 (IGLV3) subfamily genes (Supplementary Fig. 12d).

## Discussion

Antigen recognition by immunoglobulins often depends on binding with the non-linear conformational epitopes. Accordingly, short-peptide libraries for identifying the immunodominant epitopes and antigen-binding immunoglobulins from the sera of recovered patients would be ineffective. Mimicking the exact three-dimensional structure of the epitope using a combination of linear peptides forming the antigenic surface would allow for identifying virus-specific antibodies[38]. However, this would require a high-resolution 3D structure of the antigen-antibody complex and laborious co-immobilization of the respective peptide sequences reproducing the fold of the pathogen's protein. Here, we have shown that the bacteriophage's ability to present relatively long polypeptides exceeding 200 a.a. on its surface enables the selection of virus-neutralizing antibodies as if the full-length purified protein was used as an antigen. However, it is critical to highlight that recombinant proteins produced in prokaryotes often exhibit an inaccurate 3D structure, leading to a loss of natural conformational epitopes. Another limitation of this method is the lack of proper antigen glycosylation, which is sometimes critical for antibody neutralization. However, research has demonstrated that by utilizing such antigen fragments exposed on the phage surface, it is possible to isolate highly effective neutralizing antibodies. While certain antibodies exclusively target glycosylated antigens, the most potent neutralizing antibodies, revealed in this study were obtained by enrichment with both types of antigens: rRBD with eukaryotic protein glycosylation and phage-exposed antigens produced in *E.coli*. Another limitation is that, currently, the high-throughput synthesis of oligonucleotides is restricted by 350 nt, limiting the size of the translated fragment of the protein. Therefore, complex conformational epitopes longer than 100 a.a. cannot be used in large-scale screening

**Article**

**Fig. 5 | Characterization of RBD-specific mAbs and efficiency of their enrichment by different strategies. a** Correlation of RBD binding (x-axis) of isolated mAbs with their neutralization activity (y-axis). VH-VL combinations with identical heavy chain are connected by lines. mAbs from recovered donors are colored in red, and mAbs from vaccinated donors are colored in green. Antibodies are divided into four groups according to their binding/neutralizing activity: high, moderate, mild, and weak activity. **b** Enrichment ratio of best neutralizing clones in libraries enriched under various conditions (fold change of the analyzed IgG clone percentage in the designed library/percentage of the same clone in the initial library). **c** Representative data on the percentage of the selected VH-VL combinations from the recovered donor (Vir-1) enriched under various conditions. **d** Heatmap illustrates pseudo-virus neutralizing activity for the designated mAbs from the recovered donor (Vir-1).

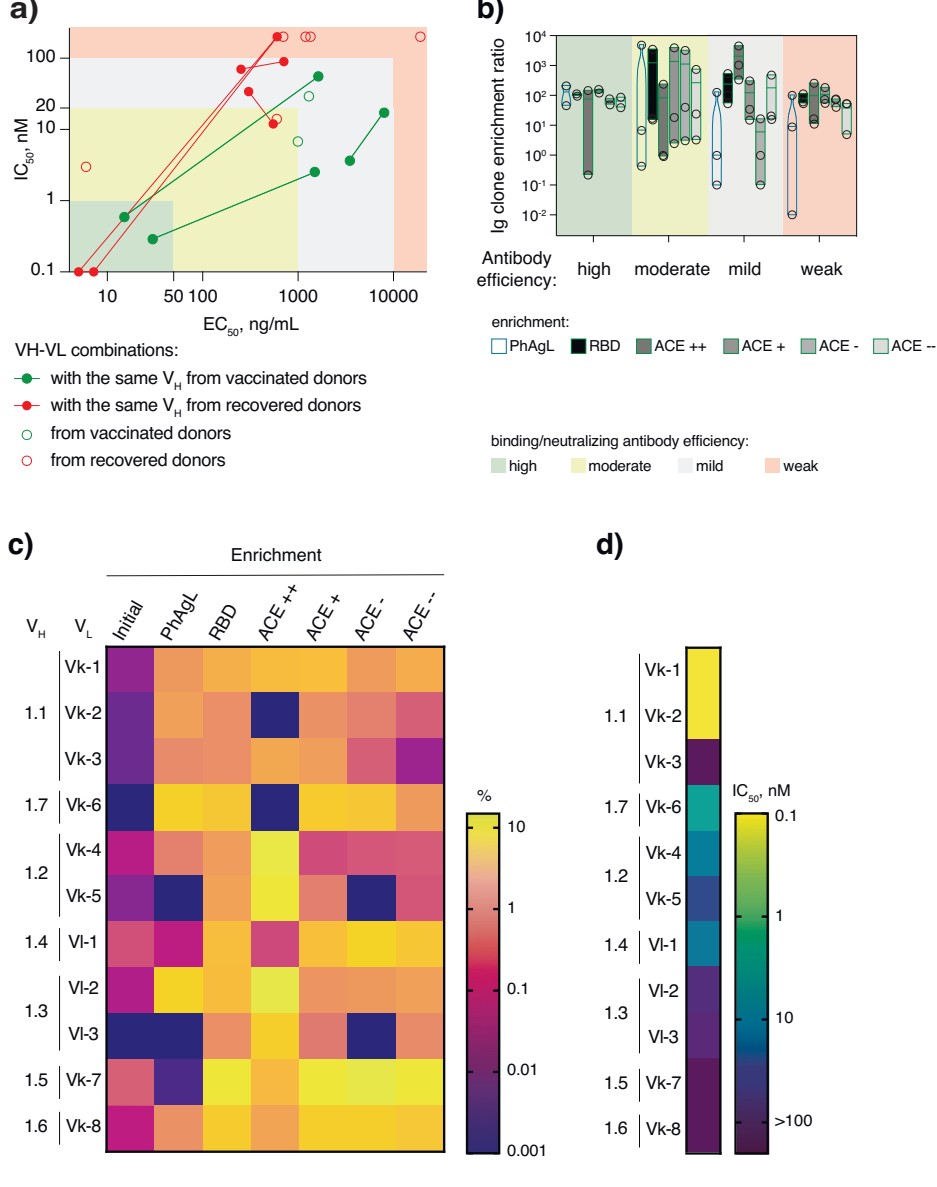

experiments. Alternatively, standard PCR amplification can be applied to construct libraries expressing proteins up to 500 a.a. long exposed on the phage surface.

Although the discovered monoclonal antibodies had been isolated from convalescent patients infected early in the pandemic (ancestral Wuhan strain) or vaccinated with the original variant of Sputnik-V[39,40], six of them effectively neutralized the immune evasive Delta strain, while two of them effectively neutralized Omicron. Although we used a small antigen library, consisting of five polypeptides belonging to one virus to prove the concept, this approach could be extended to discover antigen-specific B cells recognizing other viruses or reactive to human self-antigens. In this scenario, the development of phage libraries exposing 200–500-mer polypeptides spanning the entire viral or specific human proteome will facilitate the discovery of Ag-specific mAbs, eliminating the requirement for prior information on the target antigen structure.

Isolating antigen-specific cells from the total pool of circulating B lymphocytes, along with the subsequent combinatorial VH-VL pairing, was repeatedly shown to result in a high proportion of non-functional VH-VL combinations[41]. For really efficient isolation of high-affinity antigen-specific antibodies, along with native VH-VL pairing, IgM+ and IgD+ cells should be depleted. IgG+ B-cell enrichment was earlier shown to significantly improve the identification of target-specific clones[25]. Thus, we predict that generating lentiviral libraries with a size of about $10^6$ clones will enable the identification of a vast majority of high-affinity antigen-specific antibodies persisting in the human body. An additional advantage of this method is that it can select not only memory B cells with surface IgG, but also plasma cells, which are believed to be more mature, lack surface IgGs, and produce higher doses of antibodies.

There are currently four widely used platforms of antibody discovery: hybridoma, single B-cell cloning, scFv phage/yeast display library screening, and microfluidic-based B-cell cloning. The latter, the emerging microfluidic-based platform, enables the construction of IgG libraries with native VH-VL pairing from hundreds of thousands of B cells[42–45]. The power to enrich these libraries against hundreds or even thousands of antigens simultaneously can be of great advantage. Hence, two-dimensional screening techniques for biopanning lentiviral or yeast antibody libraries against antigen libraries, which comprise genetically encoded polypeptides exposed on the phage surface, offers the significant benefit of expeditiously identifying monoclonal antibodies with wide range of specificities. This methodology can provide a detailed description of immune response against

**Table 1 | Efficiency of the isolated antibody panel**

| Origin | Patient | ID | IGHV | IGLV/IGKV[a] | RBD Binding ELISA EC$_{50}$, ng/mL | Pseudovirus Neutralization IC$_{50}$, nM | | | Antibody efficiency |
|---|---|---|---|---|---|---|---|---|---|
| | | | | | | Wuhan | Delta | Omicron | |
| Vaccinated | Vac-1 | Vac-1.1 | IGHV1-46-01 | IGKV3-15-01 | 1300 | 30 | 30 | NA | mild |
| | | Vac-1.2 | IGHV3-30-18 | IGLV6-57-01 | 3500 | 3.8 | 40 | NA | mild |
| | | Vac-1.3 | IGHV1-46-01 | IGLV3-21-02 | 1000 | 6.5 | 30 | NA | moderate |
| | Vac-2 | Vac-2.1 | IGHV4-59-01 | IGKV3-11-01 | 30 | 0.3 | 30 | NA | high |
| | Vac-3 | Vac-3.1 | IGHV1-69-09 | IGLV3-21-02 | 15 | 0.6 | 100 | 3.4 | high |
| Recovered | Vir-1 | Vir-1.1 | IGHV3-53-01 | IGKV1-9-01 | 5 | 0.1 | 0.2 | >>100 | high |
| | | Vir-1.2 | IGHV3-30-18 | IGKV1-33-01 | 550 | 12 | 30 | >>100 | moderate |
| | | Vir-1.3 | IGHV4-61-01 | IGLV3-25-03 | 250 | 70 | >>100 | 6.6 | mild |
| | | Vir-1.4 | IGHV3-53-01 | IGLV3-21-02 | 600 | 14 | NA | NA | moderate |
| | | Vir-1.5 | IGHV2-5-01 | IGKV1-39-01 | 1200 | >>100 | NA | NA | weak |
| | | Vir-1.6 | IGHV3-66-01 | IGKV1-9-01 | 700 | >>100 | NA | NA | weak |
| | | Vir-1.7 | IGHV4-39-01 | IGKV2-28-01 | 6 | 3 | >>100 | >>100 | moderate |
| | Vir-2 | Vir-2.1 | IGHV3-53-02 | IGKV3-11-01 | >20000 | >>100 | NA | NA | no |
| | | Vir-2.2 | IGHV3-13-01 | IGKV1-39-01 | >20000 | >>100 | NA | NA | no |
| | | Vir-2. 3 | IGHV3-9-01 | IGKV1-39-01 | 1350 | >>100 | NA | NA | weak |

[a]For each VH, only VL which forms the mAb with the strongest binding to RBD is depicted.

**Table 2 | Binding kinetics (KD) of the mAbs with Wuhan-RBD, Alpha-RBD, Gamma-RBD, Delta-RBD and Omicron-RBD (BA.1) were measured by SPR**

| Antibody | Variants of SARS-CoV-2 | $K_a$ (1/Ms) | $K_d$ (1/s) | $K_D$ (M) |
|---|---|---|---|---|
| Vac-3.1 | Wuhan RBD | $3.1 \times 10^5$ | $1.3 \times 10^{-6}$ | $4.2 \times 10^{-12}$ |
| | Alpha RBD | $3.0 \times 10^5$ | $1.4 \times 10^{-6}$ | $4.7 \times 10^{-12}$ |
| | Gamma RBD | $3.0 \times 10^5$ | $1.4 \times 10^{-6}$ | $4.7 \times 10^{-12}$ |
| | Delta RBD | $1.2 \times 10^3$ | $1.00$ | $8.3 \times 10^{-4}$ |
| | Omicron RBD | $2.7 \times 10^5$ | $1.3 \times 10^{-6}$ | $4.8 \times 10^{-12}$ |
| Vir-1.7 | Wuhan RBD | $1.0 \times 10^6$ | $2.5 \times 10^{-5}$ | $2.5 \times 10^{-11}$ |
| | Alpha RBD | $0.9 \times 10^6$ | $6.0 \times 10^{-5}$ | $6.7 \times 10^{-11}$ |
| | Gamma RBD | $0.7 \times 10^6$ | $2.4 \times 10^{-4}$ | $3.4 \times 10^{-10}$ |
| | Delta RBD | $1.2 \times 10^6$ | $1.3 \times 10^{-3}$ | $1.2 \times 10^{-9}$ |
| | Omicron RBD | $2.0 \times 10^4$ | $1.2 \times 10^{-2}$ | $6.0 \times 10^{-7}$ |

a wide panel of antigens simultaneously. The proposed two-dimensional screening is suitable for evaluating the immune response in pathologies where the specific immunogen is obscure, like autoimmune diseases with unknown etiology. In this work, we have demonstrated the results of a comparative analysis of the output clones generated from primary enriched B cells. Nevertheless, combinatorial VH-VL pairing of enriched B cells results in only a fraction of functional combinations in the initial libraries (<0.1%). Moreover, we succeeded in discovering functional clones representing <0.01% in the initial libraries. Thus, we believe that two-dimensional screening of IgG libraries with natively-paired VH-VL repertoires against phage-based protein libraries will dramatically facilitate the discovery of rare antigen-specific B cells.

## Methods
### Patients and samples
All patients with moderate COVID-19 were outpatients, not requiring hospitalization. All COVID-19 subjects were confirmed positive according to quantitative reverse transcription PCR (RT-qPCR). All vaccinated donors received two doses of Gam-COVID-Vac (Sputnik-V) with a 21-day interval between the doses. None of the volunteers had experienced serious adverse events after vaccination. Before vaccination, no nucleocapsid (N)- and RBD-specific IgGs were detected in the sera of naïve individuals without COVID-19 symptoms. The study protocol was approved by the independent Ethics Committee of the Central Clinical Hospital of the Russian Academy of Sciences (protocol #143). The study was conducted according to the International Conference on Harmonization Guidelines for Good Clinical Practice and the Declaration of Helsinki 1964, along with its later amendments. Informed consent was obtained from all participating individuals. Plasma samples were spun in a centrifuge for 10 min at $1000 \times g$ in order to clarify the supernatant before use. Venous blood for FACS Sorting was collected in K2 EDTA Vacutainers (BD, USA).

### Production and biotinylation of target antigen - RBD
Synthesized DNA fragment encoding S-protein RBD of the SARS-CoV-2 virus (amino acids residues 330–528, Wuhan Hu-1 strain) in frame with interleukin-2 signal sequence for secretion, 6xHis tag for purification and C-terminal AviTag for biotin labeling was cloned into the pcDNA3.1/Hygro vector using NheI and XhoI restriction endonucleases. The resulting construct was verified by Sanger sequencing (Evrogen, Moscow, Russian Federation). RBD was expressed by transient transfection in HEK 293-F cells using FreeStyle medium (Gibco, USA) for seven days at 37 °C with 8% $CO_2$ and rotating at 135 r.p.m. The protein was purified from culture supernatants by a HiTrap Chelating column, followed by gel-filtration chromatography on a Superdex 75 10/300 GL column equilibrated with PBS according to the standard protocols. Purified protein was enzymatically labeled with biotin according to the manufacturer's instructions (Avidity LLC, USA). The biotinylation of RBD was confirmed through Western blot using HRP-conjugated streptavidin (Thermo Fisher Scientific). RBD was labeled with streptavidin-APC (Biolegend Cat# 405207), streptavidin-PE (Biolegend Cat# 405204) or streptavidin-PE-Cy7 (Biolegend Cat# 405206).

### Expression and purification of recombinant soluble human ACE2
ACE2 protein was produced in suspension HEK 293 cells following the standard protocol. Cells were grown in an Erlenmeyer flask to 1 million cells per mL, then transfected via adding a 2:1 w/w mix of PEI and plasmid DNA encoding extracellular domain of human ACE2 fused with 6xHis tag. Transfected cells were incubated for 96–120 h at 37 °C in a humidified incubator with 5% $CO_2$. The media were then harvested by consecutive centrifugation at 200, 1000, and 16,000 × g and loaded on a Ni-NTA agarose

column. After washing with 10 column volumes (CV) of starting buffer (20 mM sodium phosphate pH 7.4, 500 mM NaCl, 5 mM imidazole), the column was washed with 10 CV of wash buffer (20 mM sodium phosphate pH 7.4, 500 mM NaCl, 15 mM imidazole) and the protein was eluted with 5 CV of elution buffer (20 mM $NaH_2PO_4$ pH 7.4, 500 mM NaCl, 500 mM imidazole). ACE2-containing fractions were pooled and loaded onto a Superdex 200 column in PBS buffer. The peak corresponding to the recombinant ACE2 protein was collected, aliquoted, and stored at −20 °C.

### Quantitative determination of RBD-specific IgG in human serum by ELISA

Quantitative determination of RBD-specific IgG was performed as described previously[46]. Briefly, 100 µL of recombinant $RBD_{320-537}$ produced in CHO cells was added in PBS (1 µg/mL) to the MaxiSorp 96-well plates (Nunc, Denmark) and incubated overnight at 4 °C. Next day, 150 µL of blocking buffer (PBS, 0.05% Tween-20, 0.1% sodium caseinate) was added and incubated at room temperature for 1 h. The serum samples in the blocking buffer were prepared in three dilutions (1:10, 1:50, 1:250) and three replicates in a separate 96-well plate with low sorption capacity. Next, the serum samples (100 µL/well) were added and incubated for 30 min at 37 °C with constant shaking. After the incubation, the plate was washed five times with PBST (PBS, 0.05% Tween-20) and 100 µL of HRP-conjugated anti-human IgG antibodies (Biosan, Novosibirsk, Russian Federation, Cat. # I-3021) were then added. After 30-min incubation (37 °C, 700 rpm) and washing, 100 µl of the TMB solution was added to each well and the plate was incubated for 15 min in the dark. The enzymatic reaction was stopped by adding 10% solution of orthophosphoric acid, and optical density ($OD_{450}$) was measured on a plate spectrophotometer. The curves showing the mean $OD_{450}$ value as a function of the concentration of RBD-specific IgG in the standards (BAU/mL - binding antibody units) were plotted using the GraphPad Prism 8 software (USA). These curves were used to calculate the concentrations of RBD-specific IgG in the serum samples, and the resulting value (in BAU/mL) was multiplied by the respective dilution.

### Design and generation of the Spike phage library

To create a Phage library for the S protein of SARS-CoV-2 we used the sequence from the Wuhan Hu-1 strain (UniProt: P0DTC2). Only the ectodomain of the S protein was included, excluding the transmembrane and cytoplasmic domains: $Spike_{331-530}$ (NITNLCPFGEVFNATRFASVYA WNRKRISNCVADYSVLYNSASFSTFKCYGVSPTKLNDLCFTNVYAD SFVIRGDEVRQIAPGQTGKIADYNYKLPDDFTGCVAWNSNNLDSKI VGGNYNYLYRLFRKSNLKPFERDISTEIYQAGSTPCGVEIGFNCYFP LQSYGFQPTNGVGYQPYRVVVLSFELLHAPATVCGPKKS); $Spike_{354-519}$ (NRKRISNCVADYSVLYNSASFSTFKCYGVSPTKLNDLCFTNVYA DSFVIRGDEVRQIAPGQTGKIADYNYKLPDDFTGCVIAWNSNNLDS KVGGNYNYLYRLFRKSNLKPFERDISTEIYQAGSTPCNGVEGFNCY FPLQSYGFQPTNGVGYQPYRVVVLSFELLH); $Spike_{358-410}$ (ISNCVA-DYSVLYNSASFSTFKCYGVSPTKLNDLCFTNVYADSFVIR GDEVRQ I); $Spike_{539-583}$ (VNFNFNGLTGTGVLTESNKKFLPFQQFGRDIADTT-DAVRDPQTLE); $Spike_{795-839}$ (KDFGGFNFSQILPDPSKPSKRSFIEDL LFNKVTLADAGFIKQYGD). Sequences were optimized for uniform GC content (to reduce later bias during PCR amplification) and codon usage for expression in *E. coli*. Each sequence additionally had 14 and 12 nt adaptor sequences with NcoI and NheI restriction sites added to facilitate amplification and cloning (5′ AGCCGGCCATGGCC, 3′ TGGTGAC-GATCG). The designed library was generated by independently cloning the sequences into a fADL-1e-based phage vector (Addgene plasmid # 139441; http://n2t.net/addgene:139441; RRID:Addgene_139441), containing p3-N-terminally fused 3xFLAG epitope and serine-glycine linkers. The designed library was amplified and purified with double PEG precipitation, as we have done previously[31].

### Cell sorting

Blood samples were diluted two fold in PBS with 2 mM EDTA and layered onto Ficoll-Paque Plus (GE Healthcare) and then centrifuged at

$900 \times g$ for 40 min at room temperature. The isolated PBMC were incubated with ACK lysing buffer (0.15 M $NH_4Cl$, 10 mM $KHCO_3$, 0.1 mM $Na_2EDTA$, pH = 7.3) for complete removal of red blood cells. B cells were enriched using magnetic Dynabeads (negative selection – Invitrogen, USA) following the manufacturer's instructions with >90% purity. The obtained B cells were washed with PBS, incubated with α-CD19-PE-Cy7 (BioLegend Cat# 302215 (also 302216), RRID:AB_314245), RBD-strepavidin-PE, RBD-strepavidin-Cy5, α-CD45-APC-Cy7 (BioLegend Cat# 304014, RRID:AB_314402) and SYTOX Green dead cell stain (Invitrogen, USA) for 60 min at 4 °C in the dark. The cells were sorted directly into 1.5 mL microcentrifuge tubes containing Qiazol lysis reagent (Qiagen, Germany). Sorting was carried out using a BD FACSAria III (provided by the Lomonosov Moscow State University Development Program) and the data were analyzed using FlowJo software 9.7.5 (TreeStar, USA).

Staining of antigen-specific Jurkat cells, transduced with lentiviral scFv library, by PhAgL (bacteriophages exposing protein ligands) was performed according to the protocol described previously[31,47]. Briefly, 10 millions of Jurkat cells, transduced with lentiviral scFv library, were incubated with $4 \times 10^{13}$ particles/mL of fd bacteriophage spike library (PhAgL) in 1 mL of conjugate buffer (PBS, 0.5% BSA, 2 mM EDTA) for 1 h at 4 °C with constant gentle shaking and washed three times with PBS containing 2 mM EDTA. Thereafter the cells were incubated with fluorescent antibodies (α-FLAG-APC (BioLegend Cat# 637308 (also 637307), RRID:AB_2561497), α-Human α-IgG1 Fc-PE (SouthernBiotech Cat# 9054-09, RRID:AB_2796628), SYTOX Green) for 1 h at 4 °C with constant shaking and washed twice with PBS containing 2 mM EDTA. At each step, the cells were centrifuged at 4 °C, $300 \times g$, for 8 min and all reagents were pre-cooled to 4 °C. For staining with recombinant RBD we used α-Human α-IgG1 Fc-PE, RBD-strepavidin-PE-Cy7, RBD-strepavidin-Cy5, and SYTOX Green dead cell stain. Fluorescence intensity was measured with a Cell Sorter SH800 cytometer (Sony Biotechnology, USA). We performed 2 rounds of biopanning for the libraries from the vaccinated donors and 3 rounds of biopanning for those from the recovered donors.

### scFv-Fc lentiviral library preparation

Total RNA was isolated from the sorted B cells with the RNeasy Plus Micro Kit (Qiagen, Germany). Total complementary DNA was synthesized by reverse transcription (RT) using MMLV RT according to the manufacturer's protocol (Evrogen, Russian Federation). Variable region genes of Ig chains were amplified in separate reactions for each heavy, kappa, and lambda gene. Seminested PCR using high-fidelity DNA Q5 polymerase (New England Biolabs, USA) with a set of family-specific V gene forward and reverse primers was used (Supplementary Data 2). The PCR products for VH, Vk, and Vλ genes concentrated with AMPure XP magnetic beads (Beckman Coulter, USA), were loaded on 1.5% agarose gels. DNA fragments of the expected size (350–400 bp) were excised and purified with the Monarch Gel Extraction Kit (New England Biolabs, USA). To obtain VH-VL amplicon VH PCR products were added to Vk or Vλ PCR products at an equimolar ratio and overlapping PCR was performed. Overlapping PCR products were loaded on 1.0% agarose gels. DNA fragments of the expected size (750–800 bp) were excised and purified with the Monarch Gel Extraction Kit (New England Biolabs, USA). VH-VL amplicons with kappa and lambda chains were combined in an equimolar ratio. The resulting variable VH-VL amplicon libraries were cloned as scFv into the lentiviral vector pLV2-Fc-MTA coding for a membrane-anchored human antibody Fc fragment[32]. Jurkat cells were transduced with these viruses. The transduced Jurkat cells were analyzed by FACS to select the cells carrying antigen-specific antibodies.

To obtain VH-VL amplicons for the recovery of antigen-specific antibodies from the enriched scFv libraries, sorted Jurkat cells were used as a matrix for the subsequent PCR with primers complementary to pLV2-Fc-MTA vector backbone. The resulting variable VH-VL amplicon libraries

were used for NGS or cloned as scFv into the lentiviral vector pLV2-Fc-MTA for the subsequent library enrichment.

## Illumina library preparation and deep sequencing

The lysed sorted Jurkat cells containing enriched scFv libraries were used to prepare enriched VH-VL amplicons. To avoid skewed distribution of the enriched VH-VL sequences and perform efficient DNA amplification, we utilized emulsion PCR for DNA amplification as previously described in ref. 48 with primers, complementary to pLV2-Fc-MTA vector backbone. The obtained VH-VL amplicon was used in two separate PCR reactions to generate overlapping amplicons covering heavy and light IG chains (Supplementary Fig. 13). The overlap contains IG heavy chain CDR3 used for chain pairing after sequencing. Each 25 µL PCR reaction contained 5 ng DNA, 200 µM of each of dNTP, 1× Tersus polymerase in 1× Tersus Buffer (Evrogen, Russian Federation), and 0.2 µM of either Len-pLV2 Amp1 For (TCGTCGGCAGCGTCAGATGTGTATAAGAGACAGAGTCTTGCACTTGTCACGAAT)/ Len-pLV2 Amp1 Rev (GTCTCGTGGGCTCGGAGATGTGTATAAGAGACAGAACCGCCTCCACCTAAGC) primers to generate amplicon I or Len-pLV2 Amp2 For (TCGTCGGCAGCGTCAGATGTGTATAAGAGACAGACAAGATTTGGGCTCGCTA) primers combined with IGH multiplex primer mix (MiLaboratories, USA) to generate amplicon II. The amplification profile was: 2 min at 95 °C followed by 12 cycles of 20 s at 95 °C, 20 s at 55 °C, 40 s at 72 °C and final extension for 2 min at 72 °C. One µL of the obtained PCR product was used in the second 25 µL PCR reaction containing 200 µM of each dNTP, 1 µL of Unique Dual Indexing primer mix (Illumina, USA), and 1× Tersus polymerase in 1× Tersus Buffer amplified for 6 cycles using the same amplification profile as for the first PCR. The obtained libraries were pooled and sequenced on Illumina Miseq (paired-end 300 + 300).

## Cells and culturing conditions

The cell lines were cultured in the media supplemented with 10% fetal bovine serum (Gibco, USA), 10 mM Hepes, penicillin (100 U/mL), streptomycin (100 µg/mL), and 2 mM GlutaMAX (Gibco, USA) at 37 °C with 5% $CO_2$. HEK 293-T lentiviral packaging cell line (Clontech, USA) was cultured in Dulbecco's modified Eagle's medium (Gibco, USA). The Human Jurkat (TIB-152) cell line was obtained from the Institute of Cytology RAS culture collection (St. Petersburg, Russian Federation) and was cultured in RPMI 1640 (Gibco, USA). HEK 293-F cell line for IgG expression was cultured in Freestyle medium (Gibco, USA) without any supplementation.

## Computational analysis

The paired-end read samples of the sequenced VH-VL amplicons were filtered by the corresponding primer sequences with cutadapt v4.1[49] (—discard-untrimmed, @—action = retain). VH and VL repertoires were extracted from sequencing data using MiXCR v3.0.13[50] (mixcr amplicon module, @—OvParameters.parameters.absoluteMinScore = 40, @—OdParameters.absoluteMinScore = 25, @—OjParameters.parameters.absoluteMinScore = 40). The information about VH-VL combinations within the repertoire was gathered based on the simultaneous presence of these chains in the paired reads in the amplicon libraries containing both VH and VL sequences. These libraries were also used for D/J genes and CDR3 nucleotide sequence statistics, while the libraries with VH amplicons only – for full chain sequence and clarification of the V gene.

In order to build the Sankey and read-coverage distribution diagrams (Fig. 4a), we used the procedure for filtering the initial data. Only clones with the match number greater than 10, and light chains $V_L$ matching more than five percent of the maximum match for each individual heavy chain $V_H$ were taken into consideration.

To compare our identified SARS-CoV2-neutralizing antibodies to the published and patented antibodies, we used data from CoV-AbDab (accessed on 19.02.2024)[37]. This dataset was filtered by the following criteria: full-sized antibodies only (no nanobodies); neutralizing wild type SARS-CoV2 strain; V gene of both heavy and light chain is originated from the human locus. To visualize the distribution of V gene usage, the length of amino acid CDR3 sequences, and Ig heavy and light chains combinations over the dataset, the corresponding statistics were calculated. In the case of CDR3 length, the flanking C and W/F were taken into account. For the diagrams plotting, the library seaborn (version 0.13.0) was used.

## Human IgG construction, expression, purification

Heavy and light chains were cloned in the pFUSE-CHIg-hG1 and pFUSE2-CLIg-hK or pFUSE2-CLIg-hlambda vectors (InvivoGen, USA), respectively. The cloned insertions were verified by Sanger sequencing (Evrogen, Moscow, Russian Federation). Then purified plasmids were transfected into HEK 293-F cells and expressed in serum-free medium FreeStyle™ (Thermo Fisher Scientific, USA). PEI transfection agent (Thermo Fisher Scientific, USA) was used for transfection. Monoclonal antibodies were expressed during 4–6 days. Immunoglobulins were purified from the cultivation medium on HiTrap Protein G HP columns (Merck, Germany), followed by size exclusion chromatography using the Superdex 200 column (Cytiva, USA) according to the standard protocols. The purified immunoglobulins were quantified using the homemade Human IgG ELISA Quantitation Set and verified with Pierce BCA Protein Assay Kit (Thermo Fisher Scientific, USA), 12% denaturing SDS-PAGE under native and reducing conditions.

## Screening for antigen-specificity by ELISA

ELISAs to test for monoclonal recombinant IgG binding to RBD protein were performed as previously described in ref. 51, with some modifications. 96-well MaxiSorp plates (Nunc, Denmark) were coated with 50 µL of recombinant protein at a concentration of 2 µg/mL in carbonate buffer, with protein allowed to coat the wells overnight at 4 °C. Plates were then washed with 250 µL of PBST and blocked with 250 µL of 2% nonfat dry milk in PBS for 1 h at 37 °C. The purified antibodies were diluted in conjugate buffer (PBST with 0.5% nonfat dry milk) to concentrations in the range from 2000 to 0.2 ng/mL. 50 µL of the purified antibodies were added to each well and incubated for 1 h at 37 °C. Then the plates were washed 3 times with wash buffer. Goat anti-human anti-IgG-Fc horseradish peroxidase (HRP)-conjugated antibodies (Millipore Cat# AP113P, RRID:AB_11214132) were diluted 1:5000 in conjugate buffer, and 50 µL was added to each well. After 1 h at 37 °C, the plates were washed 5 times with wash buffer and 50 µL of TMB substrate was added. After 10 min at room temperature, the reaction was stopped with 50 µL of 10% phosphate acid, and the $OD_{450}$ was read on a VarioScan plate reader. All the isolated antibodies demonstrated no cross-reactive binding to non-relevant proteins.

For phage ELISA 96-well MaxiSorp plates were coated with 50 µL of anti-whole anti-human antibodies (Sigma–Aldrich Cat# I1886, RRID:AB_260125) at a concentration of 2 µg/mL in carbonate buffer at 4 °C overnight, then blocked and washed three times with PBST and incubated with purified mAbs at concentration of 200 ng/mL at 37 °C for 1 h. Plates were again washed 3 times with PBST, then incubated with the phage exposing respective protein at concentration of $1 \times 10^{12}$ particles/mL for 1 h at 37 °C. Then the plates were washed 5 times with wash buffer. Anti-M13-HRP antibodies (GE Healthcare Cat# 27-9420-01) were diluted 1:10,000 in conjugate buffer, and 50 µL was added to each well, incubated for 1 h at 37 °C, washed 5 times, and developed with TMB substrate as described above. Irrelevant IgG FL[32] was used as negative control mAb.

To assess the level of expressed and purified mAbs homemade Human IgG ELISA Quantitation Set was performed as described earlier[52]. Briefly 96-well MaxiSorp plates were coated with anti-whole anti-human antibodies (Sigma–Aldrich Cat# I1886, RRID:AB_260125). After subsequent blocking and washing steps different dilutions of

culture supernatants or purified mAbs were added for 1 h at 37 °C. After extensive washing step Goat anti-human anti-IgG-Fc HRP-conjugated antibodies (Millipore Cat# AP113P, RRID:AB_11214132) were added in dilution 1:5000 in conjugate buffer. The quantitative estimation of IgG was carried out using purified human IgG with determined concentration as a standard (Sigma–Aldrich Cat# I2511).

### Generation of Spike pseudotyped lentivirus
HEK 293-T cells and ACE2-overexpressed HEK 293-T cells (HEK 293T-ACE2) were cultured in advanced DMEM (Gibco, USA) supplemented with 10% fetal bovine serum (HyClone Characterized FBS, South America origin), 2 mM GlutaMAX™ Supplement (Gibco, USA), 1x Antibiotic-Antimycotic (Gibco, USA) at 37 °C with 8% $CO_2$.

The human immunodeficiency virus (HIV-1)-based lentiviral packaging system was used for pseudovirus production. HEK 293-T cells were grown in T75 flasks to 60-75% confluence and co-transfected with two packaging plasmids (15 μg of pMDLg_pRRE (GAG), Addgene ID: 12251; 5 μg of pRSV_Rev, Addgene ID: 12253), a transfer plasmid containing the reporter gene, firefly luciferase (Fluc) (15 μg of pCDH-CMV-Luc2P per T75 flask) and a plasmid encoding Wuhan Hu-1 strain SARS-CoV-2 S protein with a deletion of C-terminal 19 amino acid residues (2 μg of pcDNA3.1-Spike_del19 per T75 flask) using PEI (75 μg per T75 flask) (Polysciences, Inc., USA) according to the manufacture's instruction. 72 h post-transfection, culture supernatants containing pseudoviruses were harvested, centrifuged at $4000 \times g$ for 20 min to remove cell debris, then filtered with a syringe filter (0.45-μm pore size) and stored at −80 °C in 3-mL aliquots until use.

### SARS-CoV-2 pseudovirus neutralization assay
Pseudovirus neutralization assay was performed as previously described in ref. 53. The 96-well plates were seeded at $2 \times 10^4$ cells/well with HEK 293T-ACE2 cells. After 24 h of incubation, the culture supernatant was aspirated gently to leave 45 μL in each well. Then, 5 μL of antibody serial dilutions were added to each well, followed by infection of HEK 293T-ACE2 with pseudoviruses (50 μL per well). 3-fold serial dilutions of antibody samples were made in triplicate with the beginning dilution of 100 nM in a final volume of 100 μL. 3 wells without the addition of antibodies and 3 wells without the addition of pseudoviral particles served as the virus controls and cell controls respectively.

After 48 h, the culture supernatant was aspirated. Cells were lysed with 100 μL lysis buffer (25 mM Tris-PO₄, pH 7.8, 1% Triton X-100, 10% glycerol, 2 mM DTT, 2 mM EDTA) for 5 min at room temperature; 80 μL aliquots of cell lysate were transferred to 96-well black-walled plates, followed by addition of 20 μL luciferase substrate (Bright-Glo™ Reagent, Promega, USA). Luciferase activity was measured using the Thermo Scientific Varioskan Flash. The data were processed using Origin Software. The half-maximal neutralizing concentration ($IC_{50}$) of the antibodies was defined as an antibody concentration at which the relative light units (RLUs) were reduced by 50% compared with the virus control wells after subtraction of the background RLUs in the control groups with cell controls.

### Surface plasmon resonance (SPR)
SPR experiments were performed using the Biacore T200 system (GE Healthcare, USA). In brief, experiments were performed at 25 °C in HBS-EP + buffer (Cytiva, USA). The RBD was immobilized onto CM5 chip (GE Healthcare, USA) in 0.01 M sodium acetate buffer pH 5.0 using standard EDC/NHS protocol (Cytiva, USA). For competitive binding experiments, RBD-modified chip was saturated with 100 nM of the first antibody followed by an injection of 100 nM of the second antibody or 2 mM of soluble ACE2 at a flow rate of 20 μL/min.

### Statistics and reproducibility
The statistical significance of Ag binding between enriched Ig libraries by FACS was determined using two-tailed unpaired t-test.

### Reporting summary
Further information on research design is available in the Nature Portfolio Reporting Summary linked to this article.

### Data availability
Data supporting the findings of this work are available within the paper and in the Supplementary files. Raw BCR-seq data have been deposited to the Sequence Read Archive (SRA) with the accession number PRJNA1117008. The source data behind the graphs in the paper can be found in Supplementary Data 3.

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

## Acknowledgements

This study was supported by the Russian Science Foundation, grant # 23-44-00043. This research was performed within the framework of the Creation of Experimental Laboratories in the Natural Sciences Program and Basic Research Program at HSE University. H.Z. received financial support from the National Natural Science Foundation of China (grant number: 82261138553) and the National Key Research and Development Plan of China (grant number: 2018YFE0200400).

## Author contributions

Y.A.L.: Conceptualization, Methodology, Formal analysis, Investigation, Data curation, Writing – original draft preparation, review and editing, Supervision, Project administration; S.S.T. Formal analysis, Investigation, Writing – original draft preparation, review and editing; L.A.O., S.S.D., I.Y., I.M., A.S., T.G., I.E.E. I.N.F., Y.A.M., V.A., N.N.K., M.A.S.: Methodology, Investigation; N.L.A. and A.I.A.: Blood sample acquisition, Patient data management; M.P.R., T.V.B., V.M.B., H.Z., and I.V.S.: provided study operations support; Y.P.R: Writing – original draft preparation, review and editing; A.G.G: Writing – original draft preparation, Funding acquisition, Supervision. All authors reviewed, edited, and approved the final manuscript.

## Competing interests

The authors declare no competing interests.
