## [Peer Review File · Communications Biology]

Reviewers' comments:

Reviewer #1 (Remarks to the Author):

The authors describe discovery of potent anti-SARS-CoV-2 antibodies from patient sera, using an original approach based on (1) isolation of antigen-positive cells and incorporation of the antibody-coding regions into lentivirus-encoded library expressed in mammalian cells, (2) bait antigen expressed on phage, in RBD-form and (3) bait antigen expressed as a library of immunodominant Spike subunits of different lengths. Strongly binding, and importantly, strongly neutralizing antibodies were isolated. The analysis of the initial antibody library, as well as the subpopulations identified in the selection procedure is monitored with NGS analysis, and the final potentially neutralizing antibodies are characterized for their germline identity.

This work employs a broad spectrum of methodology, the manuscript is well-conceived and interesting to read. At the same time, a large amount of data is presented, and can be challenging to the reader to draw own conclusions, and here additional support could be offered:

(1) On which basis were the subunits of Spike chosen to be presented in the library? (apart of the comment of immunodominance)

(2) Why were additional segments of Spike not included – to possibly deliver novel information on relevant epitopes not identified in the past reports?

(3) Could the advantage of the screening on genetically encoded fragments in phage be put in the limelight, even in the perspective? This is the important “2nd dimension” featured in the title and the abstract.

(4) Please include the basic biophysical analysis and the results of binding studies of the final antibody candidates (to comment on Page 8).

Please find below a list of remarks which I hope you will find helpful.

Page 3, byline to Figure 1: “...(shaded red). While PhAgL...” - this should be one sentence

Page 4: “randomly paired VH-VL libraries” – would you reconsider the comments on the beneficial effects of correct chain pairing in the Introduction section regarding the chains in your library are randomly paired? Maybe include few positive reports on antibodies isolated from such libraries.

Page 4: “were expressed as dimers on the plasma membrane” – do you think since many viruses infect one cell, heterodimers are also possible?

Page 6. Figure 4a, lower row, last panel: is this a control experiment to visualize how much reactivity is there with RBD with library clones? Please explain.

Figure S1: I recommend including the percentage numbers in all quadrants, at least in the last plot, where the ACE-affected categories are presented. Further, the legend describes the RBD:ACE2 ratio 10:1, which is not presented in the Figure – please explain or amend.

Page 7, byline to Figure 5: “The size of the node is equal to the sum of the flows connected to it.

The data filtration procedures are provided in Supplementary materials“ – please refer to the specific section of supplementary materials. Is that passage in the Materials and Methods section?

Page 7. Figure 5b: no SD is shown, as promised in the byline

Page 8: “EC50“, 50 in subscript

Page 8: „No significant correlation was found between RBD binding and pseudovirus neutralization” – this sentence should be reworded so that it is clear that it refers to RBD-epitope targeting and not

binding strength

Table 1: The title is not optimal, maybe something like “efficiency of the isolated antibody panel; the results are shown for Enriched VH with most appropriate (effective) VL and it should be explained how that was determined (best IC50)?

Figure 6C is mentioned in text previously to 6B.

Page 9, legend to Figure 6: percentage of the selected VH-VL combinations?

Page 9, byline to Figure 6: only results from one donor are presented, and there are 5 libraries available from different donors – please include other data, even if in the Supplementary section.

Page 10: “are consistent with published data” – please describe in short words and include relevant references.

Page 10: “the observed CDR3 length was shorter than that reported before” – please shortly describe and include the reference.

Page 10: “The selected clonotypes“- would you consider to reword, clonotypes are the Abs resulting from the same clones after maturation

Page 12: “CO₂“ – 2 in subscript

Page 12: “Next day, 150 µl“

Page 13: Source of “ACK lysing buffer”

Page 13: sytox green with capital letter

Page 13: commercially available antibodies should be identifiable using RRIDs (or if not available, catalogue numbers)

Page 14: “in the amplicon libraries containing VH both VL sequences“ – word order

Page 14: were expressed for 4 - 6 days

Page 14: “purification on Superdex 200 column” – please reword into gel filtration or size exclusion chromatography

Page 15: the source of Human IgG ELISA Quantitation Set?

Page 15: the source of Goat anti-human anti-IgG-Fc horseradish peroxidase (HRP)-conjugated antibodies and ID number?

Page 15: OD₄₅₀, 450 in subscript

Page 15: in the description of SDS-PAGE: surely these are reducing, and not reduced conditions?

Reviewer #2 (Remarks to the Author):

This manuscript by Lomakin and colleagues describes a phage display technology for screening human immunoglobulin responses to specific pathogen associated antigens. Using SARS-CoV-2 as a model pathogen, the authors designed phage displaying domains from the viral spike protein S and screened cells expressing single-chain antibodies derived from the cells of patients who were either infected with SARS-CoV-2 or received the Sputnik vaccine. Results of the phage screening were compared to a more conventional approach, in which antibodies were screened using full-length Spike receptor binding domain (RBD). Using both techniques, the authors were able to successfully isolate heavy/light chain combinations that bound to RBD and, in some cases, had neutralizing activity. Although many of the components of this manuscript have been described

previously (phage display, lentiviral Ig libraries, isolation of human mAbs against SARS-CoV-2, etc.), from a methods perspective this manuscript certainly describes a novel approach to antibody discovery.

Major Comments:

1. RBD expression in E. Coli can be problematic, and folding issues of bacterial-expressed protein or domains are well described in the literature. Can the authors provide additional evidence that the spike fragments displayed on fd phage are properly folded? For example, do recombinant phage react with well-defined mAbs?
2. It is somewhat surprising that several antibodies listed in Table 1 have the ability to neutralize Omicron variants? Can the authors provide any additional information on these antibodies, such as epitope specificity?
3. The authors make the claim that the phage-based approach is similarly effective at identifying antibodies as the RBD-based probe. However, in Table 1 it is unclear which heavy/light chain combinations were identified by phage and which by rRBD?
4. It would be useful to compare/contrast the described method to other approaches for identifying specific antibodies, such as single-cell RNA-Seq.

Minor Comments:

1. In the intro, please delete the sentence: “As a matter of fact, the virus-specific antibodies are more effective against SARS-CoV-2 than the T cells”. I don’t think that there really is any point in comparing the relative value of antibodies versus T cell responses, especially since both are likely valuable in different ways.
2. Figure 3 could be in the supplement.
3. The authors state that the protocol could be used without knowing much about the target antigen (from the Intro: “In this paper, we present a comprehensive platform to produce high-performance therapeutic antibodies from an antibody library, that does not require information on target antigen structure”). However, that is clearly not the case in this manuscript. Information about the target antigen structure was used to design the specific spike protein fragments displayed on phage. The authors should either explain how one might design the phage library in the absence of structural information or delete this sentence.

Reviewer #3 (Remarks to the Author):

This manuscript aims to describe a new antibody discovery approach using phage displayed antigen (S protein) peptide fragment library and Jurkat displayed scFv-Fc library via lenti-transduction. The authors claim this method is non-time-consuming and does not need structural information about the pathogenic immunogens. The overall study is complete but the methodology development (as the main point of this paper) is less impressive, for the following reasons:

1. Starting from Ag-positive sera of recovered and vaccinated donors, screening and identification of mAb clones in a few weeks / less than one month, is at par compared to current technologies –

definitely not “non-time-consuming” as claimed.

2. Phage displayed antigen peptide fragments (30-500aa) has two concerns – limited to linear but not conformational epitopes; lack of proper glycosylation sometime critical for antibody neutralization.

3. In the pre-screening step, full-length biotinylated S protein has already been used to screen B cells before library construction. Not sure why not continue use it for antibody screening.

4. The random VH-VL pairing is less impressive and certainly not state-of-the-art.

Therefore, from bio-technology perspective, I feel it is hard to see very meaningful breakthrough in this study.

First, we would like to thank reviewers for their careful reading and thoughtful suggestions. Hopefully, our response to the raised issues will significantly improve the manuscript and its scientific value. Appropriate changes were made in the main text, figures and supplemental materials according to reviewer's suggestions. All modifications in the manuscript file, are indicated by using Track Changes. Below you can find our point-by-point response to the reviewer's concerns. Reviewer's comments are in bold font, our response is in regular font.

Reviewer #1 (Remarks to the Author):

The authors describe discovery of potent anti-SARS-CoV-2 antibodies from patient sera, using an original approach based on (1) isolation of antigen-positive cells and incorporation of the antibody-coding regions into lentivirus-encoded library expressed in mammalian cells, (2) bait antigen expressed on phage, in RBD-form and (3) bait antigen expressed as a library of immunodominant Spike subunits of different lengths. Strongly binding, and importantly, strongly neutralizing antibodies were isolated. The analysis of the initial antibody library, as well as the subpopulations identified in the selection procedure is monitored with NGS analysis, and the final potentially neutralizing antibodies are characterized for their germline identity.

This work employs a broad spectrum of methodology, the manuscript is well-conceived and interesting to read. At the same time, a large amount of data is presented, and can be challenging to the reader to draw own conclusions, and here additional support could be offered:

1. On which basis were the subunits of Spike chosen to be presented in the library? (apart of the comment of immunodominance)

We thank the reviewer for their favorable summary and critical reading of our manuscript. F1(N331-S530), the longest spike fragment analyzed, was chosen due to the fact that most of the identified monoclonal SARS-CoV-2-neutralizing antibodies target conformational epitopes located within this receptor-binding domain (N331-V524). Additional spike fragments were included to explore the potential for identifying neutralizing antibodies that recognize shorter regions of RBD or even linear epitopes outside RBD:

- F2 (N354-H519) was chosen due to the existence of monoclonal SARS-CoV-2/SARS-CoV neutralizing antibodies recognizing conserved epitopes residues in the range Y369-H519 [A highly conserved cryptic epitope in the receptor binding domains of SARS-CoV-2 and SARS-CoV, Science, 2020]

- F3 (I358-I410) was included because it was assumed that S375-N394 epitope (evolutionarily highly conserved in the SARS-CoV S protein) is more likely to be both T and B cell linear target and potentially induces non-neutralizing antibodies [Mining of epitopes on spike protein of SARS-CoV-2 from COVID-19 patients, Cell Research, 2020].

- F4 (V539-E583) and F5 (K795-D839) spike fragments were chosen because it was shown that antibodies targeting immunodominant regions T553-A570 and K809-K825 located outside receptor-binding domain significantly alter virus neutralization capacities [Two linear epitopes on the SARS-CoV-2 spike protein that elicit neutralising antibodies in COVID-19 patients, Nature Communications, 2020].

In response to Reviewer's comment, we have added the following sentences to the Results section, paragraph SARS-CoV-2 spike protein phage library (lines 231-240):

"Most of the identified monoclonal SARS-CoV-2-neutralizing antibodies target conformational epitopes located within this receptor-binding domain (N331-V524). We designed a small Phage Antigen Library (PhAgL) containing full-size RBD (Spike₃₃₁₋₅₃₀) and its immunodominant fragments: Spike₃₅₄₋₅₁₉, Spike₃₅₈₋₄₁₀, Spike₅₃₉₋₅₈₃, Spike₇₉₅₋₈₃₉ (Figure 2). Spike₃₅₄₋₅₁₉ was chosen due to the existence of monoclonal SARS-CoV-2/SARS-CoV neutralizing antibodies recognizing conserved epitopes in the range of residues Y369-

H519²⁸. Spike₅₃₉₋₅₈₃ and Spike₇₉₅₋₈₃₉ were selected as it was previously shown that antibodies targeting immunodominant regions T553-A570 and K809-K825 located outside receptor-binding domain significantly alter virus neutralization capacities ²⁹. Spike₃₅₈₋₄₁₀ was included as a linear antigen potentially inducing non-neutralizing antibodies ³⁰. Oligonucleotide sequences encoding the corresponding polypeptides were cloned into fd bacteriophage in frame with p3 coat protein and 3xFlag epitope ³¹.”

2. Why were additional segments of Spike not included – to possibly deliver novel information on relevant epitopes not identified in the past reports?

We fully agree that constructing a library of known epitopes could be considered a possible bias in the present study. However, since SARS-CoV-2 is one of the most studied viruses, at least 12916 SARS-CoV-2 binding monoclonal antibodies are present in the database CoV-AbDab (The Coronavirus Antibody Database). Multiple epitopes of neutralizing antibodies are well-known to the date. Fragments used in our study are focused on S-protein fragments having hot spots in linear epitopes of potent neutralizing antibodies or immunodominant regions. Therefore, as a proof of concept we have focused on identifying potentially neutralizing antibodies to validate our Ag-specific antibody screening approach.

3. Could the advantage of the screening on genetically encoded fragments in phage be put in the limelight, even in the perspective? This is the important “2nd dimension” featured in the title and the abstract.

In response to Reviewer’s comment, we have added the following sentences to the discussion to address the issue and place more emphasis on the '2nd dimension' of our screening technology (lines 530-536):

“Hence, two-dimensional screening techniques for biopanning lentiviral or yeast antibody libraries against antigen libraries, which comprise genetically encoded polypeptides exposed on the phage surface, offers the significant benefit of expeditiously identifying monoclonal antibodies with wide range of specificities. This methodology can provide a detailed description of immune response against a wide panel of antigens simultaneously. The proposed two-dimensional screening is suitable for evaluating the immune response in pathologies where the specific immunogen is obscure, like autoimmune diseases with unknown etiology.”

4. Please include the basic biophysical analysis and the results of binding studies of the final antibody candidates (to comment on Page 8).

Thank you for this suggestion. The biophysical analysis of recombinant monoclonal antibodies was performed using the following assays: ELISA (**Figure S5**) and SARS-CoV-2 pseudovirus neutralization assay. It should be noted that requested data can be found in **Table 1** and corresponding parts of main text describing major results. In accordance with the Reviewer’s comment, we have measured the binding of two final antibodies (Vir-1.7 and Vac-3.1) with the RBD of different SARS-CoV-2 strains using SPR (**Table 2**). Additionally, we analyzed competition of these antibodies with ACE2 and previously published antibody for RBD-binding sites (**Figure S9**).

Please find below a list of remarks which I hope you will find helpful.

1. Page 3, byline to Figure 1: “...(shaded red). While PhAgL...” - this should be one sentence

We apologize for this mistake. The text has been corrected.

2. Page 4: “randomly paired VH-VL libraries” – would you reconsider the comments on the beneficial effects of correct chain pairing in the Introduction section regarding the chains in your library are randomly paired? Maybe include few positive reports on antibodies isolated from such libraries.

In accordance with Reviewer’s comment, we have included reports on isolation of antigen-specific monoclonal antibodies from randomly paired VH-VL libraries (lines 102-112):
“Despite the great number of antigen-specific and even therapeutic mAbs obtained after enriching combinatorial paired VH-VL libraries¹⁹⁻²⁴, it has been demonstrated that the native chain pairing of antibodies from immunized sources can facilitate the identification of a higher number of functional clones²⁵..... On the other hand, a promising approach involves screening public light chains that have been paired with antigen-specific VH, or employing yeast Fab display to randomly combine the aforementioned VH with donor VL²⁷...”

3. Page 4: “were expressed as dimers on the plasma membrane” – do you think since many viruses infect one cell, heterodimers are also possible?

We appreciate the reviewer’s comment. The ability to infect a single cell with more than one viral particle and therefore to express scFv-Fc as heterodimers on the cell plasma membrane has been employed in the selection of a functional bispecific heterodimeric antibodies (Selection of antibodies that regulate phenotype from intracellular combinatorial antibody libraries. PNAS 2012, PMID: 23019357). Therefore, a high multiplicity of infection is crucial for the formation of heterodimers in transduced cells. In this work we stress out the necessity to set up transduction with lentiviral libraries under conditions which mostly guarantee infection of each cell with single viral particle. Therefore, we focused on calculating viral titers that favor single virus per cell infection and used calculated titers in library transduction experiment (please, see **Figure S1**). This approach should allow screening of cells expressing one type of VH-VL on the plasma membrane.

4. Page 6. Figure 4a, lower row, last panel: is this a control experiment to visualize how much reactivity is there with RBD with library clones? Please explain.

Yes, lower row, last (far right) panel represents control experiment on binding of recombinant RBD with scFv library enriched with phages (PhAgL – Phage Antigen Library). We added the following sentence in the Figure capture (now it is **Figure 3a**) (lines 325-326):

“...Right far panels represent the control binding with recombinant RBD in the final scFv libraries, which were enriched using two different methods...”

5. Figure S1: I recommend including the percentage numbers in all quadrants, at least in the last plot, where the ACE-affected categories are presented. Further, the legend describes the RBD:ACE2 ratio 10:1, which is not presented in the Figure – please explain or amend.

Following Reviewer’s comment, we included the percentage numbers in all quadrants (now it is **Figure S2**). Additionally, we included the percentage numbers of sorted ACE-affected populations in Figure caption. RBD:ACE ratio 10:1 is presented on the far-left panel in the **Figure S2**.

6. Page 7, byline to Figure 5: “The size of the node is equal to the sum of the flows connected to it. The data filtration procedures are provided in Supplementary

materials“ – please refer to the specific section of supplementary materials. Is that passage in the Materials and Methods section?

Thank you for careful review of the manuscript. We have revised Figure legend (now it is **Figure 4**). Now it reads as follows:

“The size of the node is proportional to the sum of the flows connected to it. The data filtration procedures are provided in chapter Computational analysis of Methods section.”

7. Page 7. Figure 5b: no SD is shown, as promised in the byline

In **Figure 4b** (previously 5b), mean values displayed as bold color line and SD as transparent shaded area. To make it clear, we added the following sentence to figure capture:

“Mean represented by a line, SD depicted as a shaded area”.

In order to clarify the distinction between libraries, we compared the number of top VH-VL combinations (**Figure 4b**). The figure analyzing the number of top VH clones in different Ab libraries has been moved to the Supplement (**Figure S4**). We have added the missing information to the legend of Figure S4

8. Page 8: “EC50“, 50 in subscript

Has been edited.

9. Page 8: „No significant correlation was found between RBD binding and pseudovirus neutralization” – this sentence should be reworded so that it is clear that it refers to RBD-epitope targeting and not binding strength

In accordance with Reviewer’s comment, we have rewritten this phrase (lines 391-394):

*“Weak correlation was observed between the efficiency of RBD binding (EC_{50}) and pseudovirus neutralization (IC_{50}) by isolated mAbs (**Figure S7 A, B**). Hence, epitope specificity, rather than the affinity of the antibody’s itself, has a more pronounced impact on the efficacy of neutralizing antibodies.”*

10. Table 1: The title is not optimal, maybe something like “efficiency of the isolated antibody panel; the results are shown for Enriched VH with most appropriate (effective) VL and it should be explained how that was determined (best IC_{50})?”

The title of **Table 1** has been corrected. Thank you for pointing to this error.

In **Table 1** only one VH-VL combination with best EC_{50} for each identified VH is depicted. An explanation has been added to the end of **Table 1**: *“For each VH, only VL which forms the mAb with the strongest binding to RBD is depicted,”*.

11. Figure 6C is mentioned in text previously to 6B.

Has been edited.

12. Page 9, legend to Figure 6: percentage of the selected VH-VL combinations?

We have corrected the legend and redrawn the figure to make it clearer (now it is **Figure 5**).

13. Page 9, byline to Figure 6: only results from one donor are presented, and there are 5 libraries available from different donors – please include other data, even if in the Supplementary section.

Thank you for the suggestions. All data on characterization of enrichment efficiency by different strategies for other libraries has been added in the Supplementary section (**Figure S8**).

14. Page 10: “are consistent with published data” – please describe in short words and include relevant references.

Antibodies were compared with previously published and patented antibodies from CoV-AbDab database (accessed on 19.02.2024) [Matthew I J Raybould, Aleksandr Kovaltsuk, Claire Marks, Charlotte M Deane, CoV-AbDab: the coronavirus antibody database, Bioinformatics, Volume 37, Issue 5, March 2021, Pages 734–735]. We have added a reference to the main text and rephrased paragraph (lines 443-459...469-471):

*“Discovered antibodies had similar features with the previously described SARS-CoV2-neutralizing IgGs from both vaccinated and recovered donors³⁷: (i) VH-VL germline pairs (**Figure S11**), of our antibodies correspond to the pairs identified previously (i.e none of our antibodies have a VH-VL pair that was not described before); (ii) CDR3 length (**Figure S12 A**) of IGH of our antibodies matches with the IGH CDR3 length distribution of described antibodies; (iii) germline distribution (**Figure S12 C**) are consistent with published data in terms of frequency of V-gene usage..... Among the neutralizing antibodies, the observed CDR3 length of IGK/IGL was shorter than that reported before³⁷ but this could be due to the limited number of discovered light chains.”*

Additionally, we have added a description to the Methods (Computational Analysis) (lines 738-745):

“To compare our identified SARS-CoV2-neutralizing antibodies to the published and patented antibodies, we used data from CoV-AbDab (accessed on 19.02.2024)³⁷. This dataset was filtered by the following criteria: full-sized antibodies only (no nanobodies); neutralizing wild type SARS-CoV2 strain; V gene of both heavy and light chain is originated from human locus. To visualize the distribution of V gene usage, the length of amino acid CDR3 sequences, and Ig heavy and light chains combinations over the dataset, the corresponding statistics were calculated. In case of CDR3 length, the flanking C and W/F were taken into account. For the diagrams plotting, the library seaborn (version 0.13.0) was used.”

15. Page 10: “the observed CDR3 length was shorter than that reported before” – please shortly describe and include the reference.

Light chains were also compared with previously published antibodies from the same database, mentioned above (CoV-AbDab database). We have corrected the sentence, now it reads as follows (lines 469-471):

“Among the neutralizing antibodies, the observed CDR3 length of IGK/IGL was shorter than that reported before³⁷ but this could be due to the limited number of discovered light chains.”

16. Page 10: “The selected clonotypes“- would you consider to reword, clonotypes are the Abs resulting from the same clones after maturation

This has been corrected. Now it reads as follows (lines 465-466):

“The selected antibodies also have 8 variants of kappa chain and 8 variants of lambda chain”.

17. Page 12: “CO2“ – 2 in subscript

This has been corrected (line 583).

18. Page 12: “Next day, 150 µl“

This has been corrected (line 595).

19. Page 13: Source of “ACK lysing buffer”

The description of ACK buffer has been added to the Methods section (lines 637-638):
“ACK lysing buffer (0.15 M NH₄Cl, 10 mM KHCO₃, 0.1 mM Na₂EDTA, pH = 7.3)”

20. Page 13: sytox green with capital letter

This has been corrected (line 643).

21. Page 13: commercially available antibodies should be identifiable using RRIDs (or if not available, catalogue numbers)

RRIDs and catalogue numbers for commercially available antibodies have been added to the Methods section.

22. Page 14: “in the amplicon libraries containing VH both VL sequences“ – word order

This has been corrected (lines 730-731).

23. Page 14: were expressed for 4 - 6 days

This has been corrected (line 753):
“Monoclonal antibodies were expressed during 4 – 6 days.”

24. Page 14: “purification on Superdex 200 column” – please reword into gel filtration or size exclusion chromatography

This has been corrected (line 755):
“..followed by size exclusion chromatography using Superdex 200 column...”

25. Page 15: the source of Human IgG ELISA Quantitation Set?

We used home-made ELISA kit to assess concentration of expressed and purified IgGs. Additional description was added to the Methods section in “Human IgG construction, expression, purification” (lines 756-758):

“The purified immunoglobulins were quantified using the home-made Human IgG ELISA Quantitation Set and verified with Pierce BCA Protein Assay Kit (Thermo Fisher Scientific), 12% denaturing SDS-PAGE under native and reducing conditions.”

and in “Screening for antigen-specificity by ELISA” (lines 791-799):

“To assess the level of expressed and purified mAbs home-made Human IgG ELISA Quantitation Set was performed as described earlier⁵². Briefly 96 well MaxiSorp plates were coated with anti-whole anti-human antibodies (Sigma-Aldrich Cat# I1886, RRID:AB_260125). After subsequent blocking and washing steps different dilutions of culture supernatants or purified mAbs were added for 1 h at 37 °C. After extensive washing step Goat anti-human anti-IgG-Fc horseradish peroxidase (HRP)-conjugated antibodies (Millipore Cat# AP113P, RRID:AB_11214132) were added in dilution 1:5000 in conjugate buffer. The quantitative estimation of IgG was carried out using purified human IgG with determined concentration as a standard (Sigma-Aldrich Cat# I2511).”

26. Page 15: the source of Goat anti-human anti-IgG-Fc horseradish peroxidase (HRP)-conjugated antibodies and ID number?

RRIDs have been added.

27. Page 15: OD450, 450 in subscript

This has been corrected.

28. Page 15: in the description of SDS-PAGE: surely these are reducing, and not reduced conditions?

This has been corrected (line 758).

Reviewer #2 (Remarks to the Author):

This manuscript by Lomakin and colleagues describes a phage display technology for screening human immunoglobulin responses to specific pathogen associated antigens. Using SARS-CoV-2 as a model pathogen, the authors designed phage displaying domains from the viral spike protein S and screened cells expressing single-chain antibodies derived from the cells of patients who were either infected with SARS-CoV-2 or received the Sputnik vaccine. Results of the phage screening were compared to a more conventional approach, in which antibodies were screened using full-length Spike receptor binding domain (RBD). Using both techniques, the authors were able to successfully isolate heavy/light chain combinations that bound to RBD and, in some cases, had neutralizing activity. Although many of the components of this manuscript have been described previously (phage display, lentiviral Ig libraries, isolation of human mAbs against SARS-CoV-2, etc.), from a methods perspective this manuscript certainly describes a novel approach to antibody discovery.

Major Comments:

1. RBD expression in E. Coli can be problematic, and folding issues of bacterial-expressed protein or domains are well described in the literature. Can the authors provide additional evidence that the spike fragments displayed on fd phage are properly folded? For example, do recombinant phage react with well-defined mAbs?

We thank the reviewer for their favorable recommendation. We fully agree that folding issues of bacterial-expressed proteins can dramatically influence, in some cases, the binding of antigen-specific antibodies. Therefore, certain neutralizing antibodies that develop in humans during the anti-viral response may be lost upon the utilization of phage-displayed proteins. Nevertheless, we declare that, in other cases, and particularly at least for the part of SARS-CoV2-neutralizing antibodies, it is possible to enrich antigen-specific antibodies by utilization of spike fragments exposed on phage surface. 15 out of the 23 VH-VL combinations identified in the present study were significantly enriched by utilization of spike fragments displayed on fd phage (see **Figure 5 C, D; Figure S 8**). Our findings demonstrate that the top-performing neutralizing antibodies can be determined by this method (9 out of 11 neutralizing antibodies with $IC_{50} < 20$ nM were enriched with PhAgL - **Figure 5 C, D; Figure S 8; Table S3**).

We performed additional experiments to examine the binding of three full-sized SARS-CoV2-neutralizing hlgGs with recombinant phage exposing Spike₃₃₁₋₅₃₀ on its surface. All three tested SARS-CoV2-neutralizing monoclonal antibodies recognized recombinant phage exposing Spike₃₃₁₋₅₃₀ on its surface. We have added a description of this experiment in Supplemental Materials (**Figure S6**) and in the body text (lines 382-389):

*“Next, we performed phage ELISA using obtained monoclonal IgGs to validate the recognition capability of the Spike₃₃₁₋₅₃₀ fragment displayed on fd phage (**Figure S6**). Our aim was to determine if it could be recognized by antibodies other than those present in scFv-Fc format exposed on the Jurkat cell surface. Despite one of the three RBD-specific antibodies showing decreased binding with Spike₃₃₁₋₅₃₀ displayed on fd phage compared to eukaryotic rRBD, it was observed that all the tested antibodies exhibited statistically significant binding to phage Spike₃₃₁₋₅₃₀ in contrast to irrelevant phages and IgG.”*

2. It is somewhat surprising that several antibodies listed in Table 1 have the ability to neutralize Omicron variants? Can the authors provide any additional information on these antibodies, such as epitope specificity?

Mutations in SARS-CoV-2 can substantially reduce the binding affinity and neutralization of antibodies. However, a panel of antibodies with broad neutralization activity towards various SARS-CoV-2 strains, including Wuhan and Omicron, has been identified to date [LY-CoV1404 (bebtelovimab) potently neutralizes SARS-CoV-2 variants. Cell Reports 2022; A novel mAb broadly neutralizes SARS-CoV-2 VOCs in vitro and in vivo, including the Omicron variants.

Journal of medical virology 2023; Broad-spectrum humanized monoclonal neutralizing antibody against SARS-CoV-2 variants, including the Omicron variant. Front Cell Infect Microbiol 2023].

In response to the Reviewer's question, we have analyzed whether the best identified universal neutralizing antibody Vac-3.1 competes for RBD-binding sites with ACE2 and other well-known antibody P4A1 [Guo, Y., Huang, L., Zhang, G. et al. A SARS-CoV-2 neutralizing antibody with extensive Spike binding coverage and modified for optimal therapeutic outcomes. Nat Commun 12, 2623 (2021)]. We have measured the binding affinity with the RBD of different SARS-CoV-2 strains using SPR (**Table 2**). Additionally, we have investigated the binding properties of Vir-1.7 antibody, which effectively neutralizes the Wuhan strain but not the Delta and Omicron strain. According to the SPR analysis, Vac-3.1 does not compete with ACE2 or P4A1 for RBD-binding sites, while Vir-1.7 competes with ACE2 but not with P4A1 (**Figure S9**). We then aligned the sequences of Wuhan, Alpha, Gamma, Delta and Omicron variants of SARS-CoV2 Spike protein (S-protein) as well as the available structural models (7DDD for Wuhan, 7W92 for Delta, 7WK2 for Omicron variant). Multiple sequence alignment (MSA) was performed with MuscleWS algorithm (Robert C. Edgar, MUSCLE: multiple sequence alignment with high accuracy and high throughput, Nucleic Acids Research, Vol 32, I 5, P 1792–1797). All S-protein amino acid residues involved in intermolecular contacts in the RBD-ACE2 (7DQA) and RBD-P4A1 (7CJF) were listed in MSA group (**Figure S10 A**). Intermolecular contacts were identified by the 0.8 nm distance threshold.

Since Vac-3.1 antibody does not compete for S-protein binding with neither ACE2 nor P4A1 it is natural to assume that its epitope does not contain amino acids listed in any of groups "bound to ACE2", "bound to P4A1" or "bound to both" (**Figure S10 A**). Also, as it binds Wuhan, Alpha, Gamma, and Omicron variants with high affinity (**Table 2**), it is probable that the epitope is conservative. RBD sequence regions meeting these two criteria are 333-338, 340-370, 376-402, 410-413, 425-438, 441-443, 450-452, 462-471, 479-483, 508-526. However, since Vac-3.1 is much less efficient binding and neutralizing antibody for Delta strain, we assume that R residue in position 452, which is unique for Delta, contribute to the virus recognition (numbering corresponds to **Figure S10**).

Vir-1.7 antibody does not compete for S-protein binding with P4A1, but does so with ACE2. This may be due to the fact that Vir-1.7 epitope partially involves amino acids from the group "bound to ACE2" and, at the same time, not bound to P4A1. These most likely are 439-449 and 497-507 (numbering corresponds to **Figure S10**).

We have added the following paragraph to the main text (lines 401-409):

*"According to the SPR analysis, the best identified universal neutralizing antibody Vac-3.1, demonstrates picomolar affinity towards RBD of Wuhan, Alpha, Gamma, and Omicron strains, while showing little to no interaction with the Delta RBD (**Table 2**). This neutralizing antibody Vac-3.1 does not compete with ACE2 or the previously published neutralizing antibody P4A1³⁶ for RBD-binding sites (**Figure S9**). Combining these observations with the Vac3-1 antibody's capacity to neutralize the Wuhan and Omicron, but to a much lesser extent Delta strains, we can infer that Delta mutation L452R dramatically affects Vac-3.1 binding with RBD (**Figure S10**). Hence, the binding site of Vac-3.1 is presumably placed close to the L452, oppositely to the ACE2 binding site."*

3. The authors make the claim that the phage-based approach is similarly effective at identifying antibodies as the RBD-based probe. However, in Table 1 it is unclear which heavy/light chain combinations were identified by phage and which by rRBD?

In **Table 1**, all the best VH-VL combinations from each donor are listed. The percentage of VH-VL combinations according to the method of their identification is presented in **Figure 5 C** and new **Figure S8**. The best neutralizing VH-VL combinations (with $IC_{50} < 5$ nM) were identified using both methods, phage library and rRBD. Totally, 15 out of the 23 VH-VL combinations identified in the present study were significantly enriched by utilization of spike

fragments displayed on fd phage (see **Figure 5 C, D; Figure S8**). Wherein, 9 out of 11 neutralizing antibodies with $IC_{50} < 20$ nM were also enriched with PhAgL. From the VH-VL combinations designated in **Table 1**, only Vac-1.3, Vir-1.5, and Vir-1.4 were not enriched with phage-based approach. It should be noted, that all described in the present study RBD-specific VH-VL combinations can be identified by rRBD.

4. It would be useful to compare/contrast the described method to other approaches for identifying specific antibodies, such as single-cell RNA-Seq.

Thank you for this comment. We have addressed this issue in the introduction and added following description of applying scRNA-seq in the characterization of Ag-specific B cells (lines 72-82):

“After a population of Ag-specific B cells is obtained, single-cell RNA sequencing (scRNA-seq) can be effectively applied to study these cells^{9,10}. The main advantage of scRNA-seq is the ability to identify the sequences of individual B cell Receptors (BCRs) and characterize phenotypic markers of each analyzed B cell within a sample. Moreover, pre-exposure of analyzed cells with several biotinylated and barcoded Ags appended to classic scRNA-seq allow to study single Ag-specific B cells with various specificity in a high-throughput manner¹¹. Despite the promise of using scRNA-seq to identify Ag-specific Abs, the limitation in the number of analyzed cells (up to 20000 per assay), the high cost per sample and the necessity to obtain recombinant antigens in the first instance, complicates the identification of rare therapeutic Ag-specific Abs with a priori unknown specificity. Alternatively, therapeutic antibodies against most new disease-associated biomolecules can be obtained via phenotypic drug discovery approach^{12...}”

Minor Comments:

1. In the intro, please delete the sentence: “As a matter of fact, the virus-specific antibodies are more effective against SARS-CoV-2 than the T cells”. I don’t think that there really is any point in comparing the relative value of antibodies versus T cell responses, especially since both are likely valuable in different ways.

We have removed this sentence from the Introduction.

2. Figure 3 could be in the supplement.

Figure 3 has been moved to the Supplement. Now it is designated **Figure S1**.

3. The authors state that the protocol could be used without knowing much about the target antigen (from the Intro: “In this paper, we present a comprehensive platform to produce high-performance therapeutic antibodies from an antibody library, that does not require information on target antigen structure”). However, that is clearly not the case in this manuscript. Information about the target antigen structure was used to design the specific spike protein fragments displayed on phage. The authors should either explain how one might design the phage library in the absence of structural information or delete this sentence.

We have revised this statement to convey that only in future experiments the development of large phage libraries, covering hundreds of proteins, will enable the the identification of therapeutic antibodies without a *priori* unknown (pre-determined) target antigens.

We have corrected the following sentence in the Introduction section (lines 172-174):

“In this paper, we present a comprehensive platform for producing high-performance therapeutic antibodies from an antibody library, that potentially will not require pre-determined information on the target antigen structure.”

Also, we have modified the following phrase in the Discussion (lines 509-514):

“Although we used a small antigen library, consisting of five polypeptides belonging to one virus to prove the concept, this approach could be extended to discover antigen-specific B cells recognizing other viruses or reactive to human self-peptide antigens. In this scenario, the development of phage libraries exposing 200-500-mer polypeptides spanning the entire viral or specific human proteome will facilitate the discovery of Ag-specific mAbs, eliminating the requirement for prior information on the target antigen structure.”

Reviewer #3 (Remarks to the Author):

This manuscript aims to describe a new antibody discovery approach using phage displayed antigen (S protein) peptide fragment library and Jurkat displayed scFv-Fc library via lenti-transduction. The authors claim this method is non-time-consuming and does not need structural information about the pathogenic immunogens. The overall study is complete but the methodology development (as the main point of this paper) is less impressive, for the following reasons:

1. Starting from Ag-positive sera of recovered and vaccinated donors, screening and identification of mAb clones in a few weeks / less than one month, is at par compared to current technologies – definitely not “non-time-consuming” as claimed.

We thank the reviewer for critical reading of our manuscript. One of the key benefits of our proposed approach is the efficient and rapid enrichment of high-affinity antibodies even when the exact immunogenic sequence is unknown. By implementing a 2D screening of the lentiviral antibody library against the phage antigen library, it is possible to effectively uncover immunogenic sequences and identify specific monoclonal antibodies targeting these previously unidentified immunogenic proteins simultaneously. Current technologies in nucleotide library construction and NGS allow screening of Ag-specific antibodies towards a *priori* unknown immunogen from the desired pathogen in less than one month. Wherein, protein production using mammalian cells is time-consuming and costly. Biophysical and functional characterization of the identified mAbs requires the use of a protein from a eukaryotic expression system, while Ag-specific mAbs themselves can be isolated by enriching antibody libraries with protein fragments exposed on the phage surface. To ensure clarity and accuracy, we have removed the term “non-time-consuming”.

2. Phage displayed antigen peptide fragments (30-500aa) has two concerns – limited to linear but not conformational epitopes; lack of proper glycosylation sometime critical for antibody neutralization.

We fully agree with the Reviewer’s observation regarding the significant impact of the absence of conformational epitopes and adequate antigen glycosylation on the virus neutralization by certain antibodies. Notwithstanding these limitations, our research demonstrates the feasibility of isolating at least some neutralizing antibodies using antigens exposed on the phage surface. In accordance with Reviewer’s comment, we added discussion of these limitations (lines 488-500):

“One limitation of this method is the lack of proper antigen glycosylation, which is sometimes critical for antibody neutralization. However, research has demonstrated that by utilizing such antigen fragments exposed on the phage surface, it is possible to isolate highly effective neutralizing antibodies. While certain antibodies exclusively target glycosylated antigens, the most potent neutralizing antibodies, revealed in this study were obtained by enrichment with both types of antigens: rRBD with eukaryotic protein glycosylation and phage-exposed antigens produced in E.coli.”

3. In the pre-screening step, full-length biotinylated S protein has already been used to screen B cells before library construction. Not sure why not continue use it for antibody screening.

The primary focus of the article is to showcase how panels of recombinant antigens displayed on the phage surface can be effectively used for isolating antibodies through 2D screening to enrich the antibody library against the antigen library. As a proof of concept, a pre-screening step was employed with a full-length biotinylated S protein to confirm the presence of antigen-specific heavy and light chains in the obtained scFv combinatorial libraries. The results revealed that only about 1% of the obtained VH-VL combinations in the constructed scFv

lentiviral libraries were capable of binding with recombinant RBD (Round I, **Figure 3A**). From this, we can confidently affirm the validity of the core concept of isolating antigen-specific antibodies using 2D screening with an antigen library exposed on phage surface.

4. The random VH-VL pairing is less impressive and certainly not state-of-the-art.

We fully agree that utilizing antibody libraries with native VH-VL pairing significantly improves enrichment efficiency. As stated in the discussion (lines 515-517) *“Isolating antigen-specific cells from the total pool of circulating B lymphocytes, along with the subsequent combinatorial VH-VL pairing, was repeatedly shown to result in a high proportion of non-functional VH-VL combinations”*⁴³. Nevertheless, the aim of our study was to demonstrate the ability to enrich antibody libraries utilizing phage antigen libraries either instead of or in parallel with recombinant soluble antigen. As noted in the discussion (lines 540-542): *“...two-dimensional screening of IgG libraries with natively-paired VH-VL repertoires against phage-based protein libraries will dramatically facilitate the discovery of rare antigen-specific B cells”*.

Another goal of the study was to investigate whether combining various light chains with the same heavy chain would yield an optimal VH-VL combination. The findings revealed that, even though antigen-specific antibodies were generated by pairing different light chains with the same VH, only one specific VH-VL combination was most effective, highlighting the importance of precise matches. In the future, if a new unknown global epidemic will arise, we hope that enrichment of antibody libraries with natively paired VH-VL against antigen libraries exposed on phage surface will facilitate the isolation of Ag-specific antibodies.

Therefore, from bio-technology perspective, I feel it is hard to see very meaningful breakthrough in this study.

The development and characterization of Ag-specific mAbs are crucial for mAb-based therapy and global human health. Mammalian cell display, which relies on transfecting Ab-encoded DNA into mammalian cells, is one of the cutting-edge techniques that can be used to identify Ag-specific antibodies. Unfortunately, this Ab discovery technology is limited by the number of antigens against which antibody libraries can efficiently be screened simultaneously. Our approach allows for 2D screening of antibody library against the panel of antigen fragments exposed on the phage surface. This proposed technology enables the mapping of monoclonal antibody sequences to panels of diverse antigens, even when the exact immunogen is unknown.

REVIEWERS' COMMENTS:

Reviewer #1 (Remarks to the Author):

The authors have diligently corrected the manuscript and responded to all questions and comments of the reviewer. In particular, they have:

- Explained the choice of genetically encoded fragments selected to be used as baits on phage surface and highlighted the novelty and practical value of their method
- Showed additional data revealing the properties of discovered antibodies, allowing a better judgement on the efficiency of their system
- Included additional comments in the Figure legends and Table titles, as well included certain relevant references, which supports the understanding of their argumentation.

With this, I consider their manuscript has substantially improved.

Reviewer #2 (Remarks to the Author):

I appreciate the thoughtful comments in response to my review. I have no additional concerns.

Reviewer #3 (Remarks to the Author):

The limitation on lack of conformational epitopes seems not reflected in revision.

I'm ok with responses to other comments.

Reviewer #3 (Remarks to the Author):

The limitation on lack of conformational epitopes seems not reflected in revision.

I'm ok with responses to other comments.

In accordance with Reviewer's comment, we have modified the following phrase in the Discussion (lines 375-377):

“Here, we have shown that the bacteriophage’s ability to present relatively long polypeptides exceeding 200 a.a. on its surface enables the selection of virus-neutralizing antibodies as if the full-length purified protein was used as an antigen. However, it is critical to highlight that recombinant proteins produced in prokaryotes often exhibit an inaccurate three-dimensional structure, leading to a loss of natural conformational epitopes. Another limitation of this method is the lack of proper antigen glycosylation, which is sometimes critical for antibody neutralization.”